# Changes in apparent temperature and $PM_{2.5}$ around the Beijing-Tianjin megalopolis under greenhouse gas and stratospheric aerosol intervention scenarios

Jun Wang[1], John C. Moore[1,2*], Liyun Zhao[1*]

[1]State Key Laboratory of Earth Surface Processes and Resource Ecology, Faculty of Geographical Science, Beijing Normal University, Beijing 100875, China

[2]Arctic Center, University of Lapland, Rovaniemi, Finland

*Correspondence to:* zhaoliyun@bnu.edu.cn, john.moore.bnu@gmail.com

**Abstract.** Apparent temperatures (AP) and ground level aerosol pollution ($PM_{2.5}$) are important factors in human health, particularly in rapidly growing urban centres in the developing world. We quantify how changes in apparent temperature – that is a combination of 2 m air temperature, relative humidity and surface wind speed, and $PM_{2.5}$ concentrations – that depend on the same meteorological factors along with future industrial emission policy, may impact people in the greater Beijing region. Four Earth System Models (ESM) simulations of the modest greenhouse emissions RCP4.5, the "business-as-usual" RCP8.5 and the stratospheric aerosol intervention G4 geoengineering scenarios are downscaled using both a 10 km resolution dynamic model (WRF), and a statistically approach (ISIMIP). We use multiple linear regression models to simulate changes in $PM_{2.5}$ and the contributions meteorological factors make in controlling seasonal AP and $PM_{2.5}$. WRF produces warmer winters and cooler summers than does ISIMIP both now and in the future. These differences mean that estimates of numbers of days with extreme apparent temperatures vary systematically with downscaling method, as well as between climate models and scenarios. Air temperature changes dominate differences in apparent temperatures between future scenarios even more than they do at present because the reductions in humidity expected under solar geoengineering are overwhelmed by rising vapor pressure due to rising temperatures and the lower windspeeds expected in the region in all future scenarios. Compared with the 2010s, $PM_{2.5}$ concentration is projected to decrease 5.4 μg/m$^3$ in the Beijing-Tianjin province under the G4 scenario during the 2060s from the WRF downscaling, but decrease by 7.6 μg/m$^3$ using ISIMIP. The relative risk of 5 diseases decreases by 1.1%-6.7% in G4/RCP4.5/RCP8.5 using ISIMIP, but have smaller decrease (0.7%-5.2%) using WRF. Temperature and humidity differences between scenarios change the relative risk of disease from $PM_{2.5}$ such that G4 results in 1-3% higher health risks than RCP4.5. Urban centres see larger rises in extreme apparent temperatures than rural surroundings due to differences in land surface type, and since these are also the most densely populated, health impacts will be dominated by the larger rises in apparent temperatures in these urban areas.

**500 character non-technical text**

Apparent temperatures and $PM_{2.5}$ pollution depends on humidity and wind speed in addition to surface temperature and impacts human health and comfort. Apparent temperatures will reach dangerous levels more commonly in future because of water vapor pressure rises and lower expected wind speeds, but these will also drive change in $PM_{2.5}$. Solar geoengineering can reduce the frequency of extreme events significantly relative to modest, and especially "business as usual" greenhouse scenarios.

# 1. Introduction

Global mean surface temperature has increased by 0.92°C (0.68-1.17°C) during 1880-2012 (IPCC, 2021), which naturally also impacts the human living environment (Kraaijenbrink et al., 2017; Garcia et al., 2018). However, neither land surface temperature nor near-surface air temperature can adequately represent the temperature we experience. Apparent temperature (AP), that is how the temperature feels, is formulated to reflect human thermal comfort and is probably a more important indication of health than daily maximum or minimum temperatures (Fischer et al., 2013; Matthews et al., 2017; Wang et al., 2021). There are various approaches to estimating how the weather conditions affect comfort, but apparent temperature is governed by air temperature, humidity and wind speed (Steadman 1984; Steadman 1994). These are known empirically to affect human thermal comfort (Jacobs et al., 2013), and thresholds have been designed to indicate danger and health risks under extreme heat events (Ho et al., 2016). Analysis of historical apparent temperatures in China (Wu et al., 2017; Chi et al., 2018; Wang et al., 2019), Australia (Jacobs et al., 2013), and the USA (Grundstein et al., 2011) all find that apparent temperature is increasing faster than air temperature. This is due to both decreasing wind speeds and, especially to increasing vapor pressure (Song et al., 2022).

As the world warms, apparent temperature is expected to rise faster than air temperatures in the future (Li et al., 2018; Song et al., 2022). Hence, humans, and other species, will face more heat-related stress but less cold-related environmental stress in the warmer future (Wang et al., 2018; Zhu et al., 2019). Since most of the population is now urban, the conditions in cities will determine how tolerable are future climates for much of humanity, while the differences in thermal comfort between urbanized and rural regions will be a factor in driving urbanization. Reliable estimates of future urban temperatures and their rural surroundings require methods to improve on standard climate model resolution to adequately represent the different land surface types; especially the rapid and accelerating changes in land cover in the huge urban areas characteristic of sprawling developments in the developing world. This is usually done with either statistical or dynamic downscaling approaches, and in this article we examine both methods.

In early 2013, Beijing encountered a serious pollution incident. The concentration of $PM_{2.5}$ (particles with diameters less than or equal to 2.5 µm in the atmosphere) exceeded 500 µg/m$^3$ (Wang et al., 2014). Following this event and its expected impacts on human health (Guan et al., 2016; Fan et al., 2021) and the economy (Maji et al., 2018; Wang et al., 2020), the Beijing municipal government launched the Clean Air Action Plan in 2013. The annual mean concentration of $PM_{2.5}$ in Beijing-Tianjin-Hebei region decreased from 90.6 µg/m$^3$ in 2013 to 56.3 µg/m$^3$ in 2017, a decrease of about 38% (Zhang et al., 2019), although this is still more than double the EU air quality standard (25 µg/m$^3$) and above the Chinese FGNS (First Grand National Standard) of 35 µg/m$^3$. The concentration of $PM_{2.5}$ is related to anthropogenic emissions, but also dependent on meteorological conditions (Chen et al., 2020). Simulations suggested that 80% of the 2013-2017 lowering of $PM_{2.5}$ concentration came from emission reductions in Beijing (Chen et al. 2019). Humidity and temperature are the main meteorological factors affecting $PM_{2.5}$ concentration in Beijing in summer, while humidity and wind speed are the main factors in winter (Chen et al., 2018). Simulations driven by different RCP emission scenarios with fixed meteorology for the year 2010 suggest that $PM_{2.5}$ concentration will meet FGNS under RCP2.6, RCP4.5 and RCP8.5 in Beijing-Tianjin-Hebei after 2040 (Li et al., 2016).

The focus here is in the differences in apparent temperature and $PM_{2.5}$ that may arise from solar geoengineering (that is reduction in incoming short-wave radiation to offset longwave absorption by greenhouse gases) via stratospheric aerosol inervention (SAI), and pure greenhouse gas climates. We use all four climate models that have provided sufficient data from the G4 scenario described by the Geoengineering Model Intercomparison Project (GeoMIP). G4 specifies sulfates as the aerosol, and greenhouse gas emissions from the RCP4.5 scenario (Kravitz et al., 2011). The impacts of G4 on surface temperature and precipitation have been discussed at regional scales (Yu et al., 2015) and both are lowered relative to RCP4.5. Some studies have focused on regional impact of SAI on apparent or wet bulb temperatures: in Europe, (Jones et al., 2018); East Asia (Kim et al., 2020); and the Maritime Continent (Kuswanto et al., 2021). But none of these studies have considered apparent temperature at scales appropriate for rapidly urbanizing regions such as on the North China Plain. The only study to date on SAI impacts on $PM_{2.5}$ pollution was a coarse resolution (4°×5°) global scale model with sophisticated chemistry (Eastham et al., 2018). They simulated aerosol rainout from the stratosphere to ground level, leading to an eventual increase in ground level $PM_{2.5}$. Eastham et al. (2018) concluded that SAI changes in tropospheric and stratospheric ozone dominated $PM_{2.5}$ impacts on global mortality. However, this study included only a first-order estimate of temperature and precipitation change on PM2.5 concentration under geoengineering, and also did not consider the situation in a highly polluted urban environment such as included in our domain, and which is typical of much of the developing world.

The greater Beijing megalopolis lies in complex terrain, surrounded by hills and

mountains on three sides, and a flat plain to the southeast coast (Fig. 1). Over the period 1971-2014, apparent temperature rose at a rate of 0.42℃/10 years over Beijing-Tianjin-Hebei region, with urbanization having an effect of 0.12℃/10 years (Luo and Lau, 2021). By the end of 2019, the permanent resident population in Beijing exceeded 21 million. Tianjin, 100 km from Beijing, is the fourth largest city in China with a population of about 15 million, and Langfang (population 4 million) is about 50 km from Beijing. Thus, the region contains a comparable urbanized population as the northeast US megalopolis. Since its climate is characterized by hot and moist summer monsoon conditions, the population is at an enhanced risk as urban heat island effects lead to city temperatures warming faster than their rural counterparts.

There are large uncertainties in projecting $PM_{2.5}$ concentration in the future due to both climate and industrial policies. Statistical methods are much faster than atmospheric chemistry models (Mishra et al., 2015), and different scenarios are easy to implement. We use a Multiple Linear Regression (MLR) model to establish the links between $PM_{2.5}$ concentration, meteorology and emissions (Upadhyay et al., 2018; Tong et al., 2018). We project and compare the differences of $PM_{2.5}$ concentration under G4 and RCP4.5 scenarios, and between different $PM_{2.5}$ emission scenarios. Accurate meteorological data are crucial in simulating future apparent temperatures and $PM_{2.5}$ because all ESM suffer from bias, and this problem is especially egregious at small scales. A companion paper (Wang et al., 2022) looked at differences between downscaling methods with the same 4 Earth System Models (ESM), domain and scenarios as we use here.

In this paper, we use the downscaled data to explore the effect of SAI on apparent temperature and $PM_{2.5}$ over the greater Beijing megalopolis. The paper is organized as follows. The data and methods of calculating AP, AP thresholds, the $PM_{2.5}$ MLR model and its validation are briefly described in Section 2. The results from present day simulation and future projections on apparent temperature and $PM_{2.5}$ are given in Section 3, along with their associated impact analyses. In Section 4 we discuss and interpret the findings, and finally we conclude with a summary of the main implications of the geoengineering impacts on these two important human health indices in Section 5.

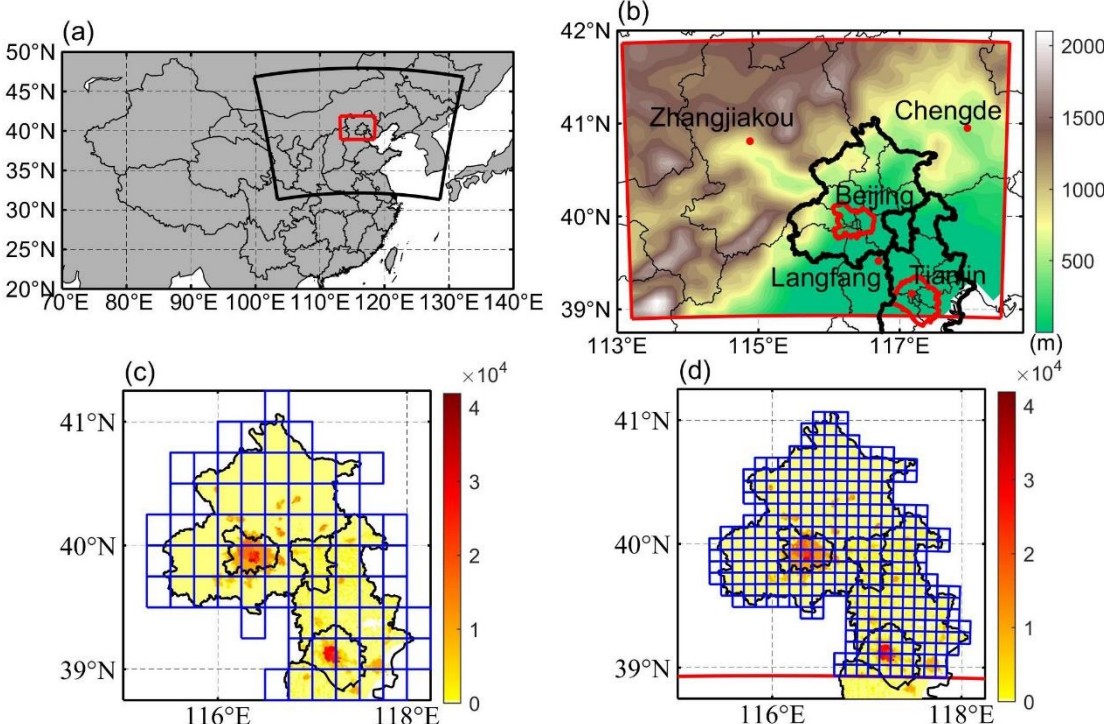

**Figure 1. a,** The 10 km WRF domain (red box) nested inside a 30 km resolution WRF domain (large black sector). **b,** The inner domain topography and major conurbations (red dots), with the urban areas of Beijing and Tianjin enclosed in red curves. Panels c and d show the population density (persons per km²) of Beijing and Tianjin provinces (defined by black borders) in 2010 and the grid cells within the Beijing-Tianjin province (blue boxes) when downscaled by ISIMIP **(c)** and WRF **(d)**.

## 2. Data and Methods

## 2.1 Scenarios, ESM, downscaling methods and bias correction

The scenarios, ESM, downscaling methods and bias correction methods we use here are as described in detail by Wang et al., (2022), and we just summarize the method briefly here. We use three different scenarios: RCP4.5 and RCP8.5 (Riahi et al., 2011) and the GeoMIP G4 scenario which span a useful range of climate scenarios: RCP4.5 is similar (Vandyck et al., 2016) to the expected trajectory of emissions under the 2015 Paris Climate Accord agreed Nationally Determined Contributions (NDCs); RCP8.5 represents a formerly business-as-usual, no climate mitigation policies, large signal to noise ratio scenario; G4 represents a similar radiative forcing as produced by the 1991 Mount Pinatubo volcanic eruption repeating every 4 years.

Climate simulations are performed by 4 ESM: BNU-ESM (Ji et al., 2014), HadGEM2-ES (Collins et al., 2011), MIROC-ESM (Watanabe et al., 2011) and MIROC-ESM-CHEM (Watanabe et al., 2011). We compare dynamical and statistical downscaling methods to convert the ESM data to scales more suited to capturing differences between contrasting rural and urban environments. To validate the downscaled AP from model results, we use the daily temperature, humidity and wind speed during 2008-2017 from the gridded observational dataset CN05.1 with the resolution of 0.25°× 0.25° based on

the observational data from more than 2400 surface meteorological stations in China, which are interpolated using the "anomaly approach" (Wu and Gao, 2013). This dataset is widely used, and has good performance relative to other reanalysis datasets over China (Zhou et al., 2016; Yang et al., 2019; Yang et al., 2023; Yang and Tang, 2023). Dynamical downscaling for the 4 ESM datasets was done with WRFv.3.9.1 with a parameter set used for urban China studies (Wang et al., 2012) in two nested domains at 30 and 10 km resolution over 2 time slices (2008-2017 and 2060-2069). We corrected the biases in WRF output using the quantile delta mapping method (QDM; Wilcke et al., 2013) with ERA5 (Hersbach et al., 2018) to preserve the mean probability density function of the output over the domain without degrading the WRF spatial pattern. All WRF results presented are after QDM bias correction. Statistical downscaling was done with the trend-preserving statistical bias-correction Inter-Sectoral Impact Model Intercomparison Project (ISIMIP) method (Hempel et al., 2013) for the raw ESM output, producing output matching the mean ERA5 observational data in the reference historical period with the same spatial resolution, while allowing the individual ESM trends in each variable to be preserved.

## 2.2 PM$_{2.5}$ concentration and emission data

In China there were few PM$_{2.5}$ monitoring stations before 2013 (Xue et al., 2021). However, aerosol optical depths produced by the Moderate Resolution Imaging Spectroradiometer (MODIS) have been used to build a daily PM$_{2.5}$ concentration dataset (ChinaHighPM2.5) at 1 km resolution from 2000 to 2018 (Wei et al., 2020). We use monthly PM$_{2.5}$ concentration data during 2008-2015 from ChinaHighPM2.5 to train the MLR model, and the data during 2016-2017 to validate it. Figure S1 shows annual PM$_{2.5}$ concentration over Beijing areas during 2008 (a) and 2017 (b).

Recent gridded monthly PM$_{2.5}$ emission data were derived from the Hemispheric Transport of Air Pollution (HTAP_V3) with a resolution of 0.1°×0.1° during 2008-2017, which is a widely used anthropogenic emission dataset (Janssens-Maenhout et al., 2015). PM$_{2.5}$ emissions over Beijing areas during 2008 (c) and 2017 (d) are shown in Fig. S1.

Future gridded monthly PM$_{2.5}$ emissions to 2050 are available in the ECLIPSE V6b database (Stohl et al., 2015), generated by the GAINS (Greenhouse gas Air pollution Interactions and Synergies) model (Klimont et al., 2017). The ECLIPSE V6b baseline emission scenario assumes that future anthropogenic emissions are consistent with those under current environmental policies, hence it is the "worst" scenario without considering any mitigation measures (Li et al., 2018; Nguyen et al., 2020). Projected emissions are shown in Fig S2, with emissions plateauing at ~40 kt/year after 2030, so we assume 2060s levels are similar. These ECLIPSE projections are significantly larger than present day estimates from HTAP_V3. We therefore estimate 2060s emissions as the recent gridded monthly PM$_{2.5}$ emissions from HTAP_V3 scaled by the ratios of 2050 ECLIPSE emission to average annual emissions between 2010 and 2015. Before

processing data, PM$_{2.5}$ concentration is bilinearly interpolated to the WRF and ISIMIP
grids, while PM$_{2.5}$ emissions are conservatively interpolated to the target grids.

## 229 2.3 Apparent temperature

We used a widely used empirical formula to calculate the apparent temperature under
shade (Steadman 1984), that combines various meteorological fields, which also has
been widely used to study heat waves, heat stress and temperature-related mortality
(Perkins and Alexander, 2013; Lyon and Barnston, 2017; Lee and Sheridan, 2018; Zhu
et al., 2021):
$$AP = -2.7 + 1.04 \times T + 2 \times P - 0.65 \times W \qquad (1)$$
where $AP$ is the apparent temperature (°C) under shade meaning that radiation is not
considered; $T$ is the 2 m temperature (°C), $W$ is the wind speed at 10 m above the ground
(m/s), and $P$ is the vapor pressure (kPa) calculated by
$$P = P_s \times RH \qquad (2)$$
where $P_s$ is the saturation vapor pressure (kPa), and $RH$ is the relative humidity (%).
$P_s$ is calculated using the Tetens empirical formula (Murray, 1966):
$$P_s = \begin{cases} 0.61078 \times e^{\left(\frac{17.2693882 \times T}{T + 237.3}\right)}, & T \geq 0 \\ 0.61078 \times e^{\left(\frac{21.8745584 \times (T-3)}{T + 265.5}\right)}, & T < 0 \end{cases} \qquad (3)$$

To assess the potential risks of heat-related exposure from apparent temperature, we
also count the number of days with $AP > 32$°C (NdAP_32) in the Beijing-Tianjin
province (Table S1). This threshold does not lead to extreme risk and death, instead it
is classified as requiring "extreme caution" by the US National Weather Service
(National Weather Service Weather Forecast Office,
*https://www.weather.gov/ama/heatindex*), but carries risks of heatstroke, cramps and
exhaustion. A threshold of 39°C is classed as "dangerous" and risks heatstroke. While
hotter AP thresholds would give a more direct estimate of health risks, the statistics of
these presently rare events mean that detecting differences between scenarios is less
reliable than using the cooler NdAP_32 threshold simply because the likelihood of rare
events are more difficult to accurately quantify than more common events that are
sampled more frequently. There is evidence that in some distributions, the likelihood
of extremes will increase more rapidly than central parts of a probability distribution,
for example large Atlantic hurricanes increasing faster than smaller ones (Grinsted et
al., 2013). But the conservative assumption is that similar differences between scenarios
would apply for higher thresholds as lower ones.

## 259 2.4 Population Data Set

Since health impacts scale with the number of people affected, we calculate the
NdAP_32 weighted by population (Fig. 1c and 1d). We employ gridded population data
(Fu et al., 2014; https://doi.org/10.3974/geodb.2014.01.06.V1) with a spatial resolution
of 1×1 km collected in 2010. The population density distribution in Beijing and Tianjin
provinces with the ISIMIP and WRF grid cells contained are shown in the Fig. 1c and
1d.

## 2.5 MLR model calibration

Many meteorological factors, such as temperature (You et al., 2017), precipitation (Guo et al., 2016), wind speed (Yin et al., 2017), radiation (Chen et al., 2017), planetary boundary layer height (Zheng et al., 2017) etc., can affect the $PM_{2.5}$ concentration. Their relative importance differs regionally. But here we consider only differences that are produced by the three scenarios, so for example we do not include precipitation in our analysis because none of the ESM simulate significant changes in our domain (Table S2). Previous studies have shown that wind and humidity are the dominant meteorological variables for $PM_{2.5}$ concentration in region we study (Chen et al., 2020), while changes in temperature and winds obviously impact local concentrations. Hence, we generate an MLR model between $PM_{2.5}$ and temperature (T), relative humidity (H), zonal wind (U), meridional wind (V) and $PM_{2.5}$ emissions (E) at every grid cell as follows:

$$PM2.5 = \sum a_i X_i + b \qquad (4)$$

Where $X_{i(i=1,2,3,4,5)}$ are the five factors, $a_i$ are the regression coefficients of the $X_i$ with $PM_{2.5},$ and $b$ is the intercept, which is a constant. We assume that all factors should be included in the regression. All the meteorological variables are from the statistical and dynamical downscaling and bias corrected results during 2008-2017, with the first 8 years used for training model and the second 2 years used for validating model. We train the MLR for the 4 ESMs under statistical and dynamical downscaling in each grid cell separately, thus accounting spatial differences in the weighting of the $X_i$ across the domain. Meteorological variables under G4, RCP4.5 and RCP8.5 during 2060-2069 are used for projection.

Here, we use $PM_{2.5}$ concentration including both primary and secondary $PM_{2.5}$ as the dependent variable and primary $PM_{2.5}$ emission and meteorological factors as independent variables in the MLR. Future $PM_{2.5}$ emissions will change in ways that are rather speculative as they depend on technological innovation and policies that are inherently unpredictable. The MLR assumes that the past emissions mix and secondary aerosols remain unchanged in the future, but meteorological factors will also indirectly impact secondary $PM_{2.5}$ to some extent.

The contributions of meteorology and $PM_{2.5}$ emissions on future concentrations are examined by using recent $PM_{2.5}$ emissions (baseline) and future $PM_{2.5}$ emissions (mitigation), and the downscaled climate scenarios. Modeled $PM_{2.5}$ concentration using recent meteorology and $PM_{2.5}$ emissions during 2008-2017 (2010s) is considered as our reference.

Collinearity of variables is inevitable in our domain. The domination of the seasonal winter and summer monsoonal weather patterns mean that temperatures, precipitation

and wind direction are all highly seasonal and correlated. In winter, precipitation is minimal and northerly winds predominate, in summer the opposite is true. These three meteorological fields are important and also important for emissions, since sources are essentially absent from the north, while temperature and humidity dominate aerosol microphysics.

We use the variance inflation factor (VIF) to test if there is excessive collinearity in our MLR models. Generally, if VIF value is greater than 10, there is collinearity problem between variables. Figure S3 shows that there are indeed collinearity problems in some areas, but not in Beijing-Tianjin province, so there is no impact on the results for the urban areas. We explored the impact of collinearity on the results in high VIF grid cells by removing factors with VIF greater than 10 and the full variables model (Fig. S4 and Fig. S5). Using ISIMIP downscaling, we only removed the temperature, while we removed the temperature and U-wind in the WRF method. $PM_{2.5}$ concentrations increased by ~1 $μg/m^2$ in all ESMs under G4 with the "baseline" scenario (Fig. S4), in contrast, $PM_{2.5}$ concentrations decreased by 5-15 $μg/m^2$ with the "mitigation" scenario (Fig. S5) after dealing the collinearity problem. This means that $PM_{2.5}$ concentration has more sensitivity to the $PM_{2.5}$ emission after accounting for collinearity. Although the absolute $PM_{2.5}$ concentrations are different accounting for collinearity, there are no significant differences in the changes of $PM_{2.5}$ concentration between G4 and the 2010s/RCP4.5/RCP8.5 in Beijing-Tianjin province.

## 2.6 MLR model validation

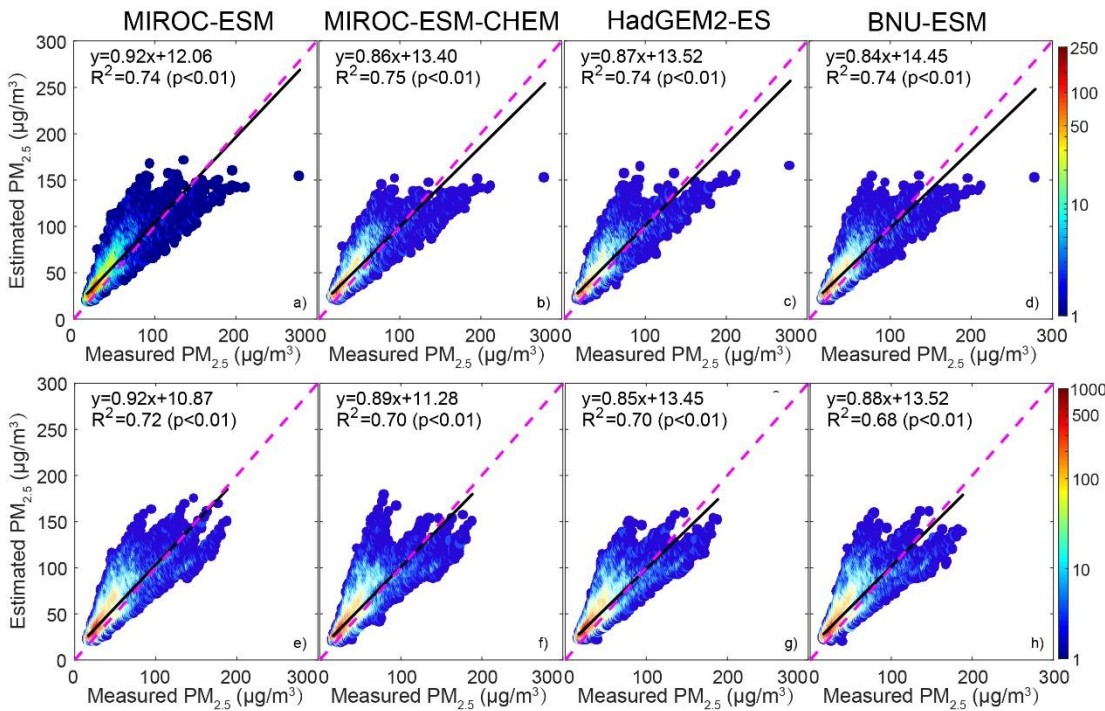

**Figure 2.** Scattergrams of $PM_{2.5}$ concentration derived by MODIS and estimated by MLR during validation period (2016-2017). Top figures **(a-d)** are the ISIMIP statistical downscaling results, and bottom figures **(e-h)** are the WRF dynamical downscaling results. $R^2$ means the variance explained

by the MLR, and color bar denotes the density of datapoints at integer intervals.

Figure 2 shows the scattergram of $PM_{2.5}$ concentration between ChinaHighPM2.5 dataset and MLR model during validation period based on ISIMIP and WRF results. Observations and MLR models have Pearson's correlations coefficients around 0.86 for ISIMIP results during the validation period, and the coefficient of determination of MLRs are 0.74-0.75 (Fig. 2a-d). WRF Pearson's correlations are slightly lower, 0.82-0.85, and explained variance ranges from 0.68-0.72 (Fig. 2e-h). These results are similar as found by Jin et al. (2022). We also compare the spatial patterns of observed and modeled $PM_{2.5}$ in Fig. S6. Both ISIMIP and WRF results can simulate the distribution characteristics of high concentration of $PM_{2.5}$ in the southeast and low concentration in the northwest.

We also tested the accuracy of our MLR model projection against simulations (Li et al., 2023) with the Community Multiscale Air Quality (CMAQ) model developed by the United States Environmental Protection Agency and which can simulate particulate matter on local scales (Foley et al., 2010; Yang et al., 2019) when coupled to WRF. We used the same meteorological forcing as Li with the "EIT1" $PM_{2.5}$ emissions scenario in 2050 under RCP4.5 (Fig.S7). The spatial patterns are well correlated in all seasons (0.68-0.73), but $PM_{2.5}$ concentrations are about twice as high in our MLR model as from Li et al., (2023). $PM_{2.5}$ concentrations from our regression model are also higher than the referenced data during 2008-2017. While the difference in absolute PM2.5 concentrations are significant, we mainly consider differences of $PM_{2.5}$ concentration between G4 and RCP4.5/RCP8.5 in our study which we cannot compare these anomalies with the single RCP4.5 scenario simulated by Li et al. (2023). We do compare the spatial pattern of differences in $PM_{2.5}$ concentration between "base" and "EIT1" under RCP4.5. Because of the small slope coefficient of $PM_{2.5}$ emission in our MLR, we do not capture the large reduction of PM2.5 concentration in the Beijing city center seen by Li et al. (2023), (Fig. S8).

## 2.7 Relative risks of mortality related to PM$_{2.5}$

We estimate the effects of $PM_{2.5}$ on mortality by considering changes in the relative risk (RR) of mortality related to $PM_{2.5}$. We lack data on mortality rates in the study domain without which we cannot estimate numbers of fatalities, just the average population-weighted RR. Burnett et al. (2014) established the integrated exposure-response functions we use. The RR is non-linear in concentration, that is an initially low $PM_{2.5}$ region will suffer higher mortality and RR than an initially high $PM_{2.5}$ region if $PM_{2.5}$ is increased by the same amount. Ran et al. (2023) provide RR values for $PM_{2.5}$ concentrations up to 200 $\mu g/m^3$ that includes the 5 main major disease endpoints (Global Burden of Disease Collaborative Network, 2013) of $PM_{2.5}$ related mortality: chronic obstructive pulmonary disease, ischemic heart disease, lung cancer, lung respiratory infection and stroke. We calculate the average population-weighted relative risks based on the gridded population dataset (Section 2.4) and $PM_{2.5}$ concentration in

the Beijing-Tianjin province defined in the Fig. 1c-1d, following Ran et al. (2023):
$$RR_{pop,k} = \frac{\sum_{g=1}^{G} POP_g \times RR_k(C_g)}{\sum_{g=1}^{G} POP_g} \quad (5)$$

$RR_{pop,k}$ is the average population-weighted relative risk of disease $k$ ($k$=1-5), $POP_g$ is
the population of gird $g$, and $RR_k(C_g)$ is the relative risk of disease $k$ when $PM_{2.5}$
concentration is $C_g$ in the grid of $g$.

## 2.8 Determination of contributions to change in AP and $PM_{2.5}$

Equation (1) describes how AP is calculated, and this can be broken down into how
much equivalent temperature is produced by each term (Fig. 3), with 2008-2017 as the
baseline interval for season-by-season contributors to AP. Across scenario seasonal
differences in contributors are then calculated as follows. We use an MLR approach,
since this minimizes the square differences from the mean across the dataset, with the
attendant assumption of independence between the data. Alternatives may also be
considered that e.g. minimize the impact of outliers by considering the magnitude of
the differences, but we prefer to keep the attractive properties of a least squares
approach. The dependent variable in the MLR is the change in AP ($\Delta AP$) and the
independent variables are changes in each factor for each future scenario,
$$\Delta AP = \sum \alpha_i X_i + \beta \quad (6)$$

where $X_{i(i=1,2,3)}$ are the daily changes of the three meteorological factors between two
scenarios: 2 m temperature ($\Delta T$), 2 m relative humidity ($\Delta RH$) and 10 m wind speed
($\Delta W$), $\alpha_i$ are the regression coefficients of the $X_i$ with $\Delta AP$, and $\beta$ is the intercept,
which is a constant. We assume that all three meteorological factors should be included
in the regression and we estimate the contributions of each factor to changes of $AP$ as:
$$K_i = \frac{\alpha_i \overline{X}_i}{\sum \alpha_i \overline{X}_i} \quad (7)$$

where $K_{i(i=1,2,3)}$ is the contributions (in units of temperature) from each factor to the
changes of the AP, and $\overline{X}_i$ are the mean differences in temperature equivalent due to
each factor between two scenarios.

The contribution of changes in each factor in changes of $PM_{2.5}$ is simpler since we
assume that the relationship between each factor and $PM_{2.5}$ is linear, and so its
contribution is the ratio of product of the regression coefficient and the change of each
factor to the change of $PM_{2.5}$.

## 3. Results

## 3.1 Recent apparent temperatures

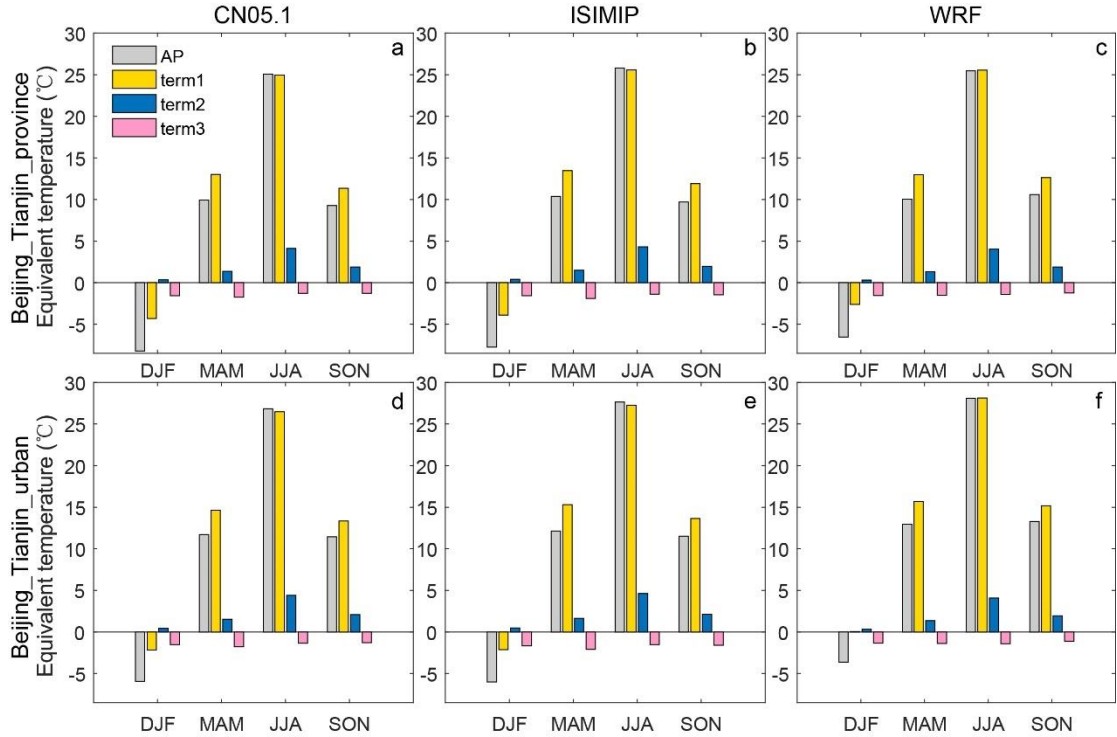


**Figure 3.** Seasonal averaged AP and equivalent temperature of each term in equation 1 for Beijing-
Tianjin province **(a-c)** and Beijing-Tianjin urban areas **(d-f)** during 2008-2017 from CN05.1 **(a, d)**, 4-
model ensemble mean after ISIMIP **(b, e)** and ensemble mean after WRF **(c, f)**. Term 1 is 1.04T, term 2
is 2P and term 3 is -0.65W.

Figure 3 shows the seasonal averaged AP and equivalent temperatures caused by
temperature, relative humidity and wind speed in Beijing-Tianjin province and Beijing-
Tianjin urban areas during 2008-2017. According to the CN05.1 results (Fig. 3a, 3d),
AP and the separate 3 terms show similar seasonal patterns over the whole province
and just the urban areas. Vapor pressure is higher in summer and wind speed is higher
in spring. AP is lower than 2 m temperature in all seasons except summer, and especially
lower in winter. AP, temperature, vapor pressure and wind speed are all higher in urban
areas than in the surrounding rural region in any season. The ISIMIP results (Fig. 3b,
3e), by design, perfectly reproduce the CN05.1 seasonal characteristics of AP,
temperature, vapor pressure and wind speed. WRF shows a similar pattern with that
from CN05.1, but for the Beijing-Tianjin province, WRF overestimates both 2 m
temperature and AP in winter by 2.1°C and by 1.7°C respectively relative to CN05.1
(Fig. 3c). In the Beijing-Tianjin urban areas, WRF overestimates the temperature and
AP relative to CN05.1 in all seasons, especially in winter (Fig. 3f).

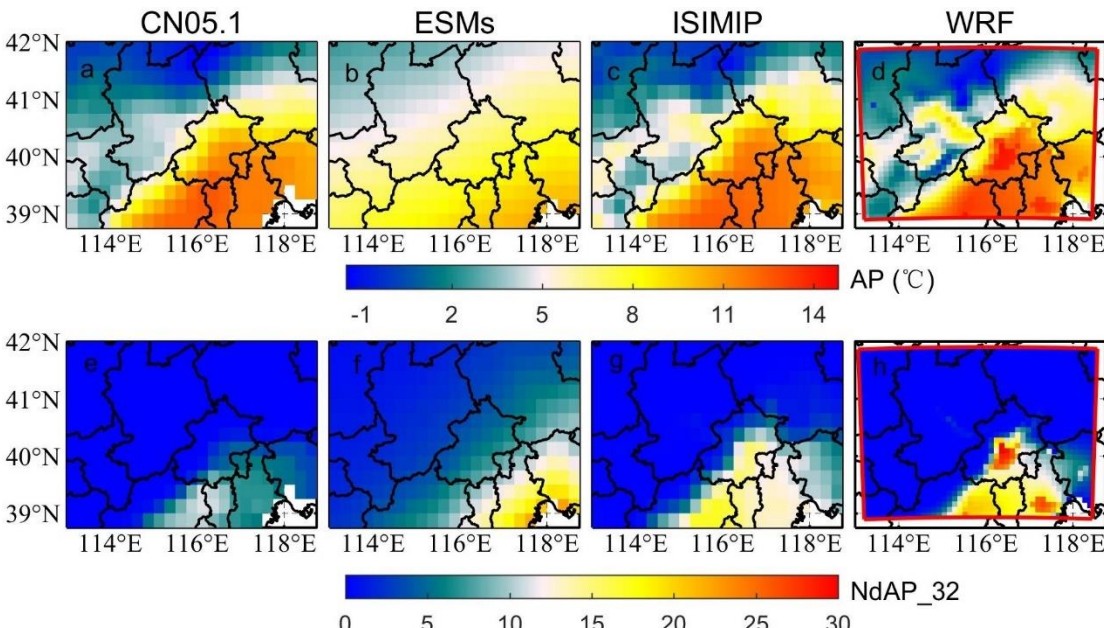

**Figure 4.** Top row: the spatial distribution of mean apparent temperature from CN05.1 **(a),** raw ESMs ensemble mean after bilinear interpolation **(b),** 4-model ensemble mean after ISIMIP **(c)** and ensemble mean after WRF **(d)** during 2008-2017. Bottom row: the spatial distribution of annual mean number of days with AP > 32℃ from CN05.1 **(e),** ESMs **(f),** ISIMIP **(e)** and WRF **(f)** during 2008-2017. Fig. S9 and Fig. S10 show the pattern of AP and NdAP_32 for the individual ESM.

We compare the simulations of mean apparent temperature and NdAP_32 from both WRF dynamical downscaling with QDM and from ISIMIP statistical downscaling during 2008-2017 in Fig. 4. Both WRF with QDM and ISIMIP methods produce a pattern of apparent temperature which is close to that from CN05.1. While the raw AP from ESMs is overestimated in Zhangjiakou high mountains and underestimated in the southern plain, and shares a similar pattern with temperature from ESMs (Wang et al., 2022). The raw ESM outputs were improved after dynamical and statistical downscaling. The average annual AP from ISIMIP (9.6-9.7°C) is 0.5°C higher than that from CN05.1 (9.1°C) over the Beijing-Tianjin province for all ESMs (Table 1). While WRF produces warmer apparent temperatures in the city centers of Beijing and Tianjin and lower ones in the high Zhangjiakou mountains than recorded in the lower resolution CN05.1 observations. There are also differences between different models after WRF downscaling. For example, apparent temperatures from the two MIROC models downscaled by WRF are the warmest. In contrast AP from all 4 ESMs after ISIMIP shows very similar patterns (Fig. S9).

ESMs tend to overestimate the number of days with AP>32℃ in southeastern Beijing and the whole Tianjin province. Both ISIMIP and WRF appear to overestimate the NdAP_32 in Beijing urban areas and the southerly lowland areas although NdAP_32 is close to zero in the colder rural areas at relatively high altitude for both downscaling methods. Some of these differences may be due to the WRF simulations being at finer resolution than the 0.25°× 0.25° CN05.1, leading to higher probabilities of high AP in urban areas (Fig. 5d). ISIMIP results also show slight overestimations, especially in the

tails of the distribution (AP>30℃) for urban areas (Fig. 5c). CN05.1 gives about 5 NdAP_32 per year in southern Beijing and Tianjin, but there are nearly 15 NdAP_32 from ISIMIP, and over 20 NdAP_32 per year from WRF downscaling in the Beijing-Tianjin urban areas during 2008-2017. NdAP_32 from WRF and ISIMIP downscaling of all ESM is overestimated relative to CN05.1. But there are differences in ESM under the two downscalings: with ISIMIP, HadGEM2-ES and BNU-ESM have more NdAP_32 than the two MIROC models, while the reverse occurs with WRF (Fig. S10).

**Table 1.** The annual mean apparent temperature and population weighted NdAP_32 in Beijing-Tianjin province and Beijing-Tianjin urban areas (Fig. 1b) from CN05.1, ISIMIP and WRF during 2008-2017.

| Data Sources | AP (℃) | | | | NdAP_32 (day yr$^{-1}$) | |
|---|---|---|---|---|---|---|
| | Provinces | | Urban | | Population weighted for province (Fig. 1c, 1d) | |
| | WRF | ISIMIP | WRF | ISIMIP | WRF | ISIMIP |
| MIROC-ESM | 10.5 | 9.6 | 13.6 | 11.4 | 22.2 | 10.1 |
| MIROC-ESM-CHEM | 10.5 | 9.6 | 13.6 | 11.4 | 21.9 | 11.0 |
| HadGEM2-ES | 9.5 | 9.6 | 12.0 | 11.4 | 12.3 | 11.1 |
| BNU-ESM | 9.4 | 9.7 | 11.8 | 11.5 | 10.2 | 12.7 |
| CN05.1 | 9.1 | | 11.1 | | 2.4 | |

The Taylor diagram of the daily mean apparent temperature in Beijing-Tianjin province and Beijing-Tianjin urban areas from 2008-2017 for the 4 ESMs shows that correlation coefficients between ESMs and CN05.1 are greater than 0.85 under both downscaling methods. Although there are differences between ESMs, the performance of WRF, with higher correlation coefficient and smaller SD (standard deviation) and RMSD (root mean standard deviation), is usually superior to ISIMIP (Fig. S11). Taking the Beijing-Tianjin urban areas as an example (Fig. S11b), under the ISIMIP method, MIROC-ESM, MIROC-ESM-CHEM and HadGEM2-ES have the same correlation coefficient (0.92) and RMSD (5.4℃) with the CN05.1, while BNU-ESM has lower correlation coefficient (0.88) and higher RMSD (7.0℃). Under WRF simulations, MIROC-ESM and MIROC-ESM-CHEM have larger correlation coefficients and smaller RMSD with CN05.1 than HadGEM2-ES and BNU-ESM.

Figure 5 shows the probability density functions (pdf) of daily AP from the four ESMs under ISIMIP and WRF in Beijing-Tianjin province and Beijing-Tianjin urban areas during 2008-2017. ISIMIP overestimates the probability of extreme cold AP relative to CN05.1 (especially BNU-ESM), although all ESM reproduce the CN05.1 pdf well at high AP. WRF can reproduce the CN05.1 distribution of AP better than ISIMIP, but high AP is overestimated relative to CN05.1 and the urban areas perform less well than the whole Beijing-Tianjin province. In urban areas all ESMs driving WRF tend to underestimate the probability of lower AP and to overestimate the probability of higher AP, especially the two MIROC models (Fig. 5d). Fig. S12 displays the annual cycle of monthly AP, with ISIMIP proving excellent by design, at reproducing the monthly AP. While under WRF downscaling AP shows more across model differences, especially during summer and with greater spread for the urban areas.

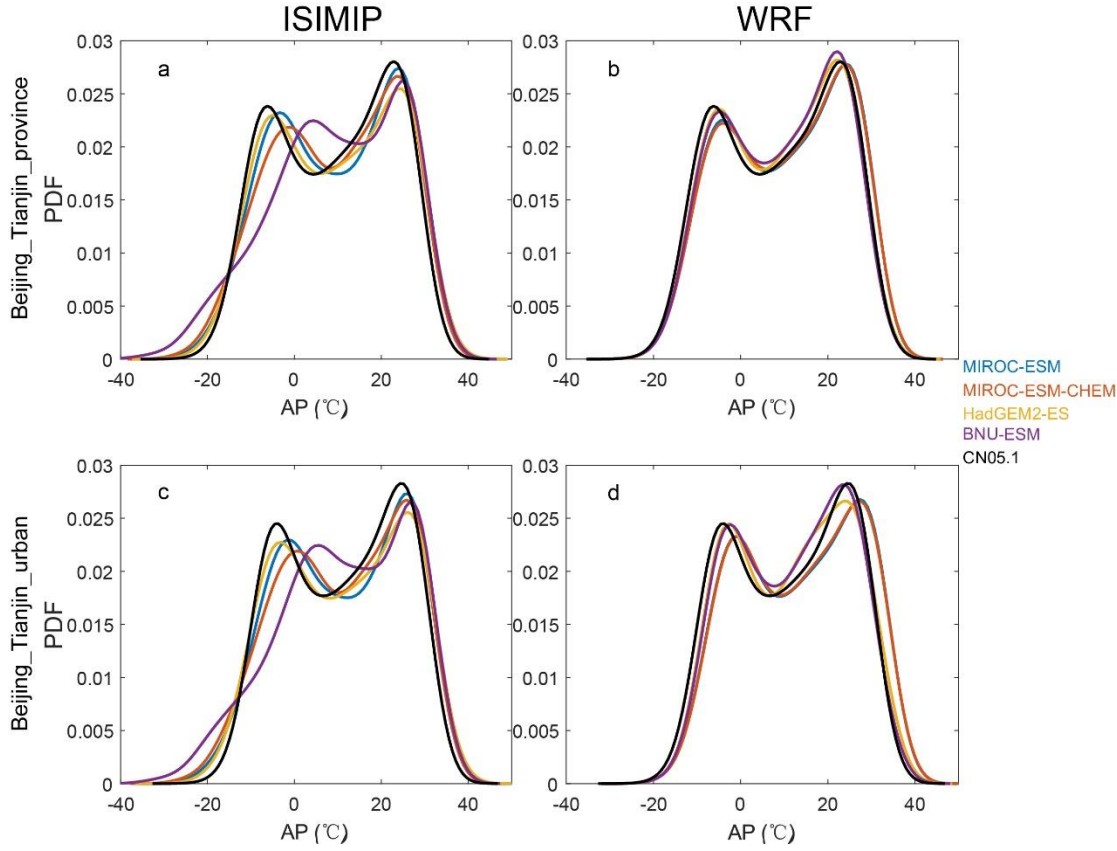

496

**Figure 5.** The probability density function (pdf) for daily apparent temperature under ISIMIP **(a, c)** and WRF **(b, d)** results in Beijing-Tianjin province **(a, b)** and Beijing-Tianjin urban areas **(c, d)** during 2008-2017.

## 3.2 2060s apparent temperatures

### 3.2.1 Changes of apparent temperature

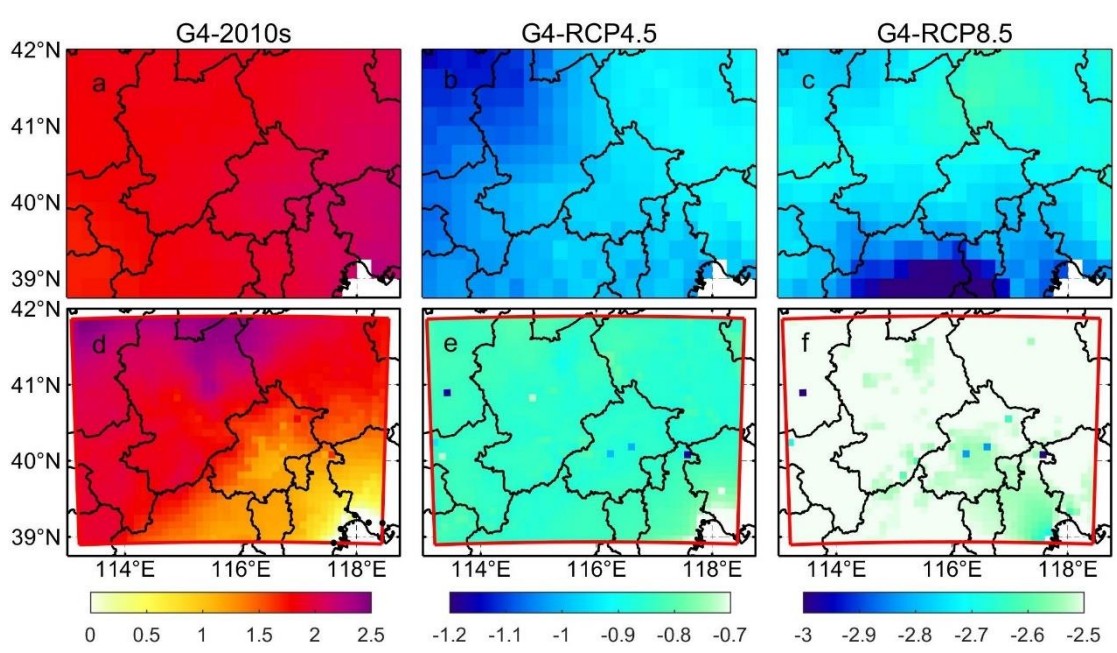


**Figure 6.** Spatial pattern of ensemble mean apparent temperature difference (℃) under different
scenarios over 2060-2069: G4-2010s (left column), G4-RCP4.5 (middle column) and G4-RCP8.5 (right
column) based on ISIMIP and WRF methods. 2010s refers to the 2008-2017 period. Stippling indicates
grid points where differences or changes are not significant at the 5% level according to the Wilcoxon
signed rank test.

Figure 6 shows the ISIMIP and WRF ensemble mean changes in the annual mean AP
under G4 during 2060-2069 relative to the past and the two future RCP scenarios.
ISIMIP-downscaled AP (Fig. 6a-6c) shows significant anomalies (p<0.05), with whole
domain rises of 2.0 ℃ in G4-2010s, and falls of 1.0 ℃ and 2.8 ℃ in G4-RCP4.5 and
G4-RCP8.5 respectively. In WRF results, AP under G4 is about 1-2 ℃ warmer than
that under 2010s, 0.8 ℃ and 2.5 ℃ colder than that under RCP4.5 and RCP8.5 over
the whole domain. Individual ESM results downscaled by ISIMIP and WRF are in Fig.
S14 and Fig. S15. For both ISIMIP and WRF downscaling results, the two MIROC
models show stronger warming than the other two models between G4 and the 2010s.
WRF-downscaled AP driven by HadGEM2-ES exhibits the strongest cooling, with
decreases of 1.7 ℃ between G4 and RCP4.5 and falls of 3.0 ℃ between G4 and RCP8.5.
Although different ESMs show different changes in AP between G4 and other scenarios,
changes in AP are almost the same everywhere for a given ESM in the ISIMIP results
(Fig. S14). WRF-downscaled AP anomalies driven by two MIROC models are larger
in the Zhangjiakou mountains and smaller in the Beijing urban areas and Tianjin city
between G4 and 2010s (Fig. S15). Changes in AP from ISIMIP results, whether across
whole province or just the urban areas, are statistically identical given scenarios (Table
2), which is consistent with patterns in figure 6. AP under G4 is 0.8 ℃ (1.0 ℃) and
2.6 ℃ (2.8 ℃) colder than that under RCP4.5 and RCP8.5 in Beijing-Tianjin urban
areas from ISIMIP (WRF) results. The warming between G4 and 2010s in urban areas
is 1.0 ℃ in WRF results, while that is 2.0 ℃ in ISIMIP results (Table 2).

**Table 2.** Difference of apparent temperature between the G4 and other scenarios for the Beijing-Tianjin
province and Beijing-Tianjin urban areas as defined in Fig. 1b during 2060-2069. Bold indicates the
differences or changes are significant at the 5% level according to the Wilcoxon signed rank test.
(Units: ℃)

| Model | G4-2010s | | | | G4-RCP4.5 | | | | G4-RCP8.5 | | | |
|---|---|---|---|---|---|---|---|---|---|---|---|---|
| | WRF | | ISIMIP | | WRF | | ISIMIP | | WRF | | ISIMIP | |
| | Urban | Province | Urban | Province | Urban | Province | Urban | Province | Urban | Province | Urban | Province |
| MIROC-ESM | **0.9** | **1.5** | **2.2** | **2.2** | **-0.5** | **-0.4** | **-0.9** | **-0.9** | **-2.3** | **-2.1** | **-2.8** | **-2.7** |
| MIROC-ESM-CHEM | **0.9** | **1.5** | **2.9** | **2.8** | **-0.4** | **-0.4** | -0.1 | -0.1 | **-2.0** | **-2.0** | **-2.1** | **-2.1** |
| HadGEM2-ES | **1.1** | **1.0** | **1.8** | **1.7** | **-1.6** | **-1.6** | **-1.6** | **-1.6** | **-3.1** | **-3.1** | **-3.3** | **-3.3** |
| BNU-ESM | **1.2** | **1.1** | **1.2** | **1.3** | **-0.8** | **-0.8** | **-1.3** | **-1.3** | **-2.8** | **-2.7** | **-2.9** | **-2.9** |
| Ensemble | **1.0** | **1.3** | **2.0** | **2.0** | **-0.8** | **-0.8** | **-1.0** | **-1.0** | **-2.6** | **-2.5** | **-2.8** | **-2.8** |


## 536 3.2.2 Contributing factors to changes in AP

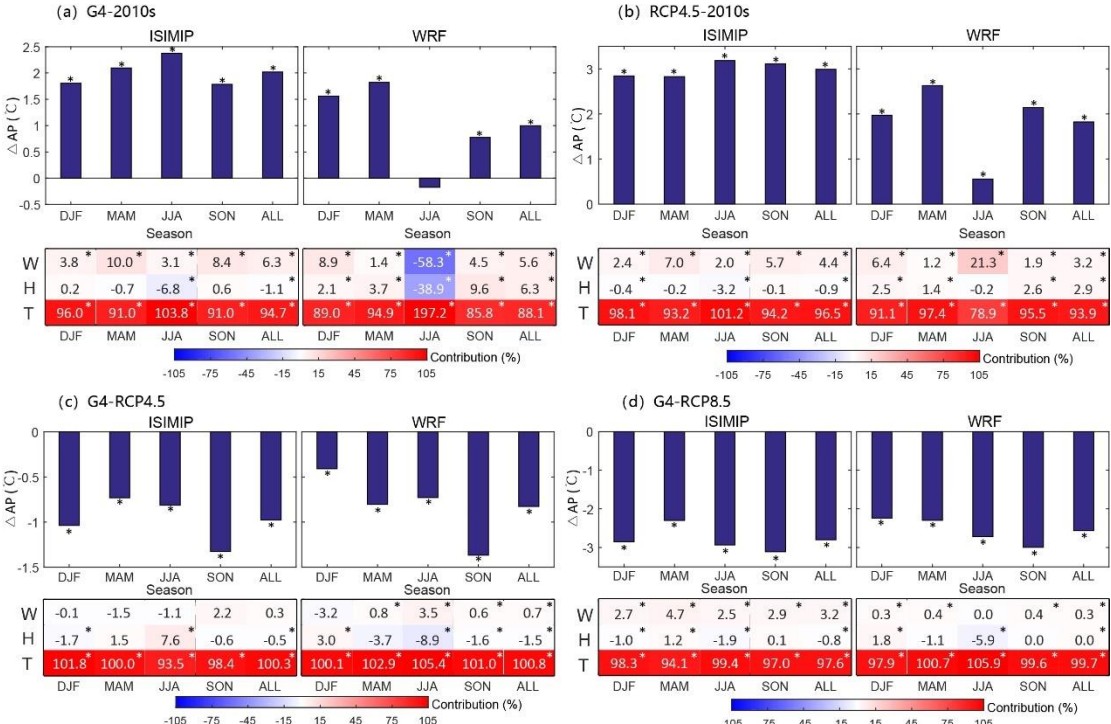

**Figure 7.** The seasonal changes of AP (ΔAP) and the seasonal contribution of climatic factors to ΔAP for Beijing and Tianjin urban areas under ISIMIP and WRF between G4 and 2010s **(a)**, G4 and 2010s **(b)**, G4 and RCP4.5 **(c)** and G4 and RCP8.5 **(d)** in the 2060s based on ensemble mean results. Colors and numbers in each cell correspond to color bar, and "*" above the columns and in the cells indicate differences are significant at the 5% significant level under the Wilcoxon test.

Figure 7 shows the ISIMIP and WRF ensemble mean changes in the annual mean AP anomalies G4 during 2060-2069 relative to the past and the two future RCP scenarios. ISIMIP-downscaled AP (Fig. 7a-7c) shows significant anomalies ($p<0.05$) across the whole domain, even for the relatively small differences in G4-RCP4.5. ΔAP by WRF is lower than that by ISIMIP. Between G4 and 2010s, AP are projected to have increases of 1.8 (1.6), 2.1 (1.8), 2.4 (-0.2), 1.8 (0.8) °C from winter to autumn in ISIMIP (WRF) results. In ISIMIP results, the contribution of temperature ranges from 91%-104%, and the contribution of wind speed ranges from 3%-10% in all seasons, while the contribution of humidity is negative or insignificant (Fig. 7a). However, the contribution of humidity is positive in WRF results (Fig. 7a). Between RCP4.5 and 2010s, annual mean AP is projected to increase by 3.0 °C and 1.8 °C in ISIMIP and WRF results respectively, which is higher than that between G4 and 2010s. The increase of temperature and decrease of wind speed have a significant impact on the annual average ΔAP contributed 97% (94%) and 4% (3%) in ISIMIP (WRF) results. The contributions of changes in humidity are significantly positive under G4 and RCP4.5 in WRF results, while it is the opposite in the ISIMIP results (Fig. 7a-7b).

Relative to RCP4.5 in the 2060s, AP is projected to decrease by 1.0 (0.4), 0.7 (0.8), 0.8 (0.7), and 1.3 (1.4) °C from winter to autumn under G4 in ISIMIP (WRF) results (Fig. 7c). In summer, the contribution from changes in temperature and humidity are 94%

(105%) and 8% (-9%) in ISIMIP (WRF) results, respectively. There are insignificant
contributions from wind speed under ISIMIP results, but a significant slight positive
contribution (0.7%-4%) under WRF results (Fig. 7c). The annual mean AP under G4 is
2.8 (2.6) ℃ lower than that under RCP8.5 in ISIMIP (WRF) result. In this case, the
contribution of changes in wind on ΔAP ranges from 3%-5% by ISIMIP, while it is
close to 0 by WRF. As expected, ΔAP is mainly determined by the changes in
temperature, with contributions usually above 90% between different scenarios.

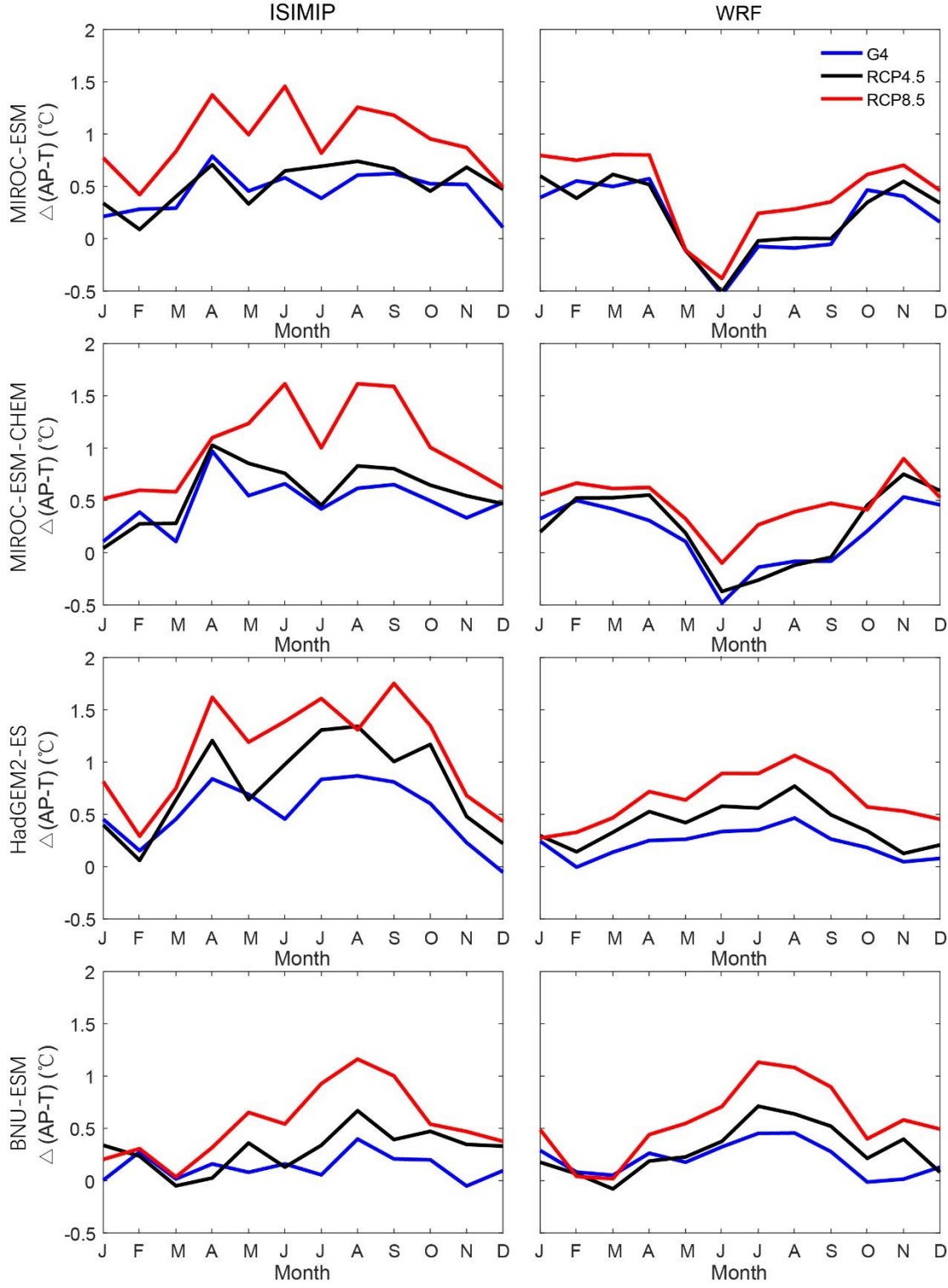


**Figure 8.** The change of apparent temperature based on air temperature under three scenarios (G4, RCP4.5 and RCP8.5) in four ESMs under ISIMIP (left column) and WRF (right column) for urban areas relative to the 2010s.

A useful measure of heat impacts that may be missed if considering only at air temperatures is the seasonality of the differences between AP and air temperature (Δ(AP-T); Fig. 8). The four model ensemble annual mean Δ(AP-T) under ISIMIP is projected to rise by 0.4℃, 0.5℃ and 0.9℃ under G4, RCP4.5 and RCP8.5, relative to the 2010s. Under WRF, Δ(AP-T) is much smaller than under ISIMIP but still rising faster than air temperatures: by 0.2℃, 0.3℃ and 0.5℃ under G4, RCP4.5 and RCP8.5 relative to the 2010s, respectively. In general, the largest anomalies in Δ(AP-T) are in summer under both WRF and ISMIP downscaling, but the two MIROC models under WRF have small or even negative Δ(AP-T) in summer with WRF.

### 3.2.3 Changes of the number of days with AP>32℃

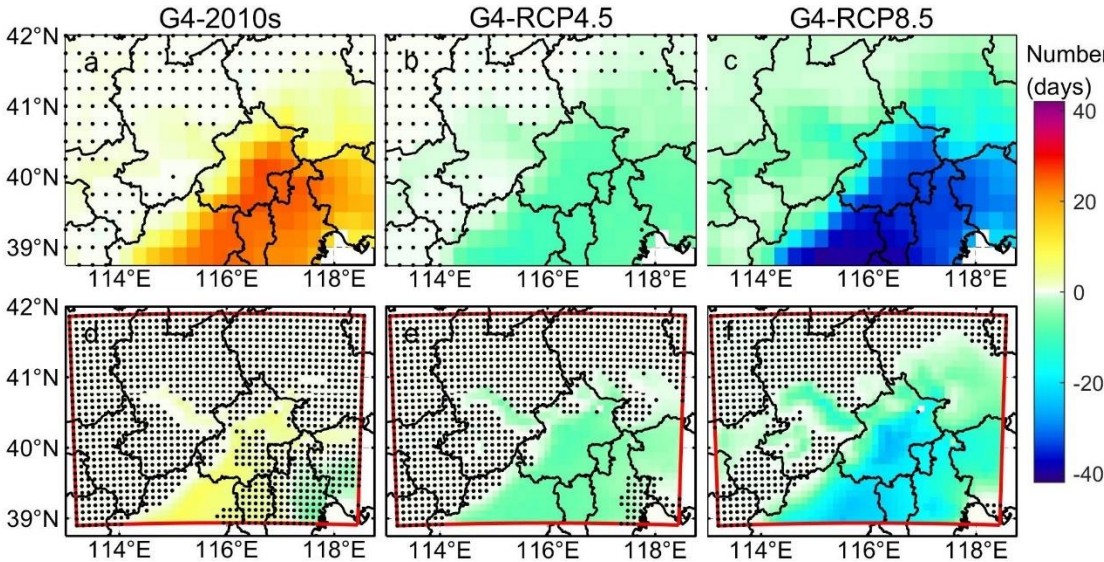

**Figure 9.** Ensemble mean differences in annual number of days with AP > 32℃ (NdAP_32) between scenarios for 2060-2069: G4-2010s (left column), G4-RCP4.5 (second column) and G4-RCP8.5 (right column) based on ISIMIP method and WRF. 2010s means the results simulated during 2008-2017. Stippling indicates grid points where differences or changes are not significant at the 5% level according to the Wilcoxon signed rank test. Corresponding ISIMIP results for each ESM are in Fig. S16, and WRF results in Fig. S17.

The NdAP_32 anomalies in Figure 9 show that ISIMIP projects an increase of about 20 days per year with AP>32 ℃ for the southeast of Beijing province and 10 days in the western areas of Beijing under G4 relative to the 2010s. NdAP_32 is about 10 days fewer under G4 than RCP4.5 with no clear spatial differences. G4 has about 35 fewer NdAP_32 days in the southern part of the domain and 20 fewer days in the western domain than the RCP8.5 scenario. In contrast WRF suggests that most areas do not show any significant difference between G4 and the 2010s, while the anomalies relative to RCP4.5 are similar as ISIMIP, the differences are insignificant over more area than

ISIMIP. G4-RCP8.5 anomalies with WRF are smaller than with ISIMIP, and differences are not significant in the Zhangjiakou high mountains. The urban areas show larger decreases in NdAP_32 than the more rural areas, even in the low altitude plain. Individual ESM show almost no statistically significant differences between G4 and RCP4.5 (Fig. S16 and S17), but the differences seen in Fig. 9 are significant because of the larger sample size in the significance test. All ESMs with ISIMIP show more NdAP_32 in the urban areas under G4 than the 2010s, while two MIROC models driving WRF show fewer NdAP_32 in Beijing-Tianjin urban areas (Fig. S16, S17).

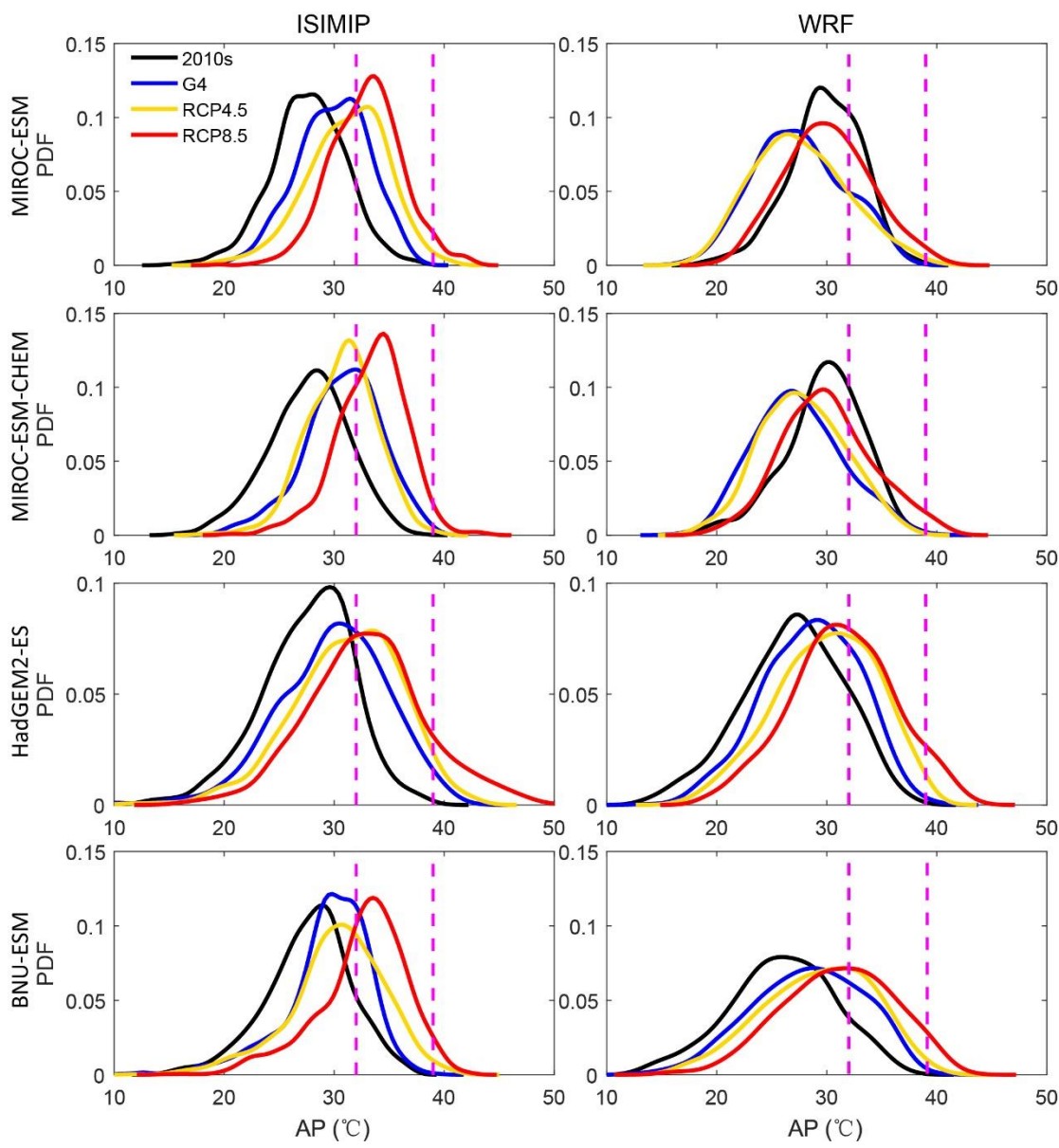

**Figure 10.** Probability density distributions of daily apparent temperature (AP) in summer (JJA) over Beijing-Tianjin urban areas under recent period (2008-2017), and the 2060s under G4, RCP4.5 and RCP8.5 scenarios from ISIMIP and WRF results. The purple dotted lines are at AP of 32°C and 39°C.

The pdf of daily apparent temperature in summer over Beijing-Tianjin urban areas (Fig. 10) shifts rightwards for G4, RCP4.5 and RCP8.5 during the 2060s relative to the 2010s.

Figure 10 shows that by the 2060s, the dangerous threshold of AP>39 is crossed frequently under RCP8.5 with both WRF and ISIMIP downscaling, but for the RCP4.5 and G4 scenarios these events are much rarer. ISIMIP results tend to show higher probability tails (extreme events) than under WRF simulations.

Population weighted NdAP_32 in the 2060s for Beijing-Tianjin province is shown in Table 3. ISIMIP downscaling suggests ensemble mean rises in NdAP_32 of 22.4 days per year under G4 relative to the 2010s, but that G4 has 8.6 and 33.5 days per year fewer than RCP4.5 and RCP8.5, respectively. NdAP_32 from WRF under G4 is reduced by 19.6 days per year relative to RCP8.5, and by 6.3 days relative to RCP4.5 (Table 3).

**Table 3.** Difference of population weighted NdAP_32 between the G4 and other scenarios for Beijing-Tianjin province (Fig. 1c, 1d) during 2060-2069. Bold indicates the changes are significant at the 5% level according to the Wilcoxon signed rank test. (Units: day $y^{-1}$).

| Beijing-Tianjin province | G4-2010s | | G4-RCP4.5 | | G4-RCP8.5 | |
|---|---|---|---|---|---|---|
| | ISIMIP | WRF | ISIMIP | WRF | ISIMIP | WRF |
| MIROC-ESM | **18.6** | -8.1 | -17.0 | 0.8 | **-35.4** | **-13.1** |
| MIROC-ESM-CHEM | **28.7** | **-10.2** | 3.9 | -2.2 | **-33.7** | **-15.5** |
| HadGEM2-ES | **25.7** | **9.4** | **-12.5** | **-13.5** | **-24.3** | **-25.3** |
| BNU-ESM | **16.4** | **13.6** | -8.6 | **-10.4** | **-40.5** | **-24.4** |
| Ensemble | **22.4±2.9** | 1.2±6.0 | **-8.6±4.5** | **-6.3±3.4** | **-33.5±3.4** | **-19.6±3.1** |

## 3.3 PM$_{2.5}$ in the 2060s

### 3.3.1 PM$_{2.5}$ scenarios in the 2060s

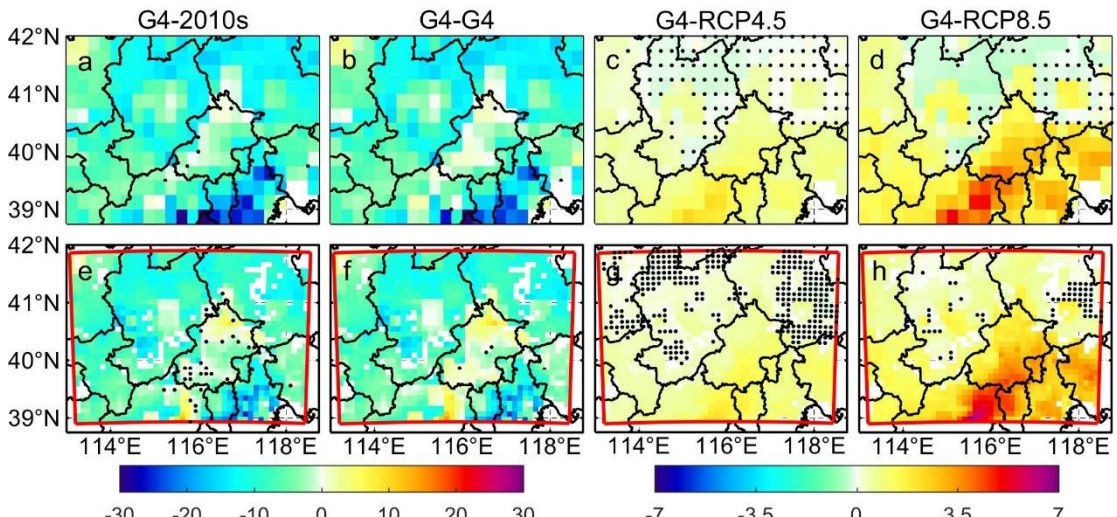

**Figure 11.** Spatial patterns of ensemble mean PM$_{2.5}$ concentration difference (μg/m$^3$) between "mitigation" under G4 in the 2060s and reference **(a, e)**, between "mitigation" and "baseline" under G4 in the 2060s **(b, f)**, between G4 and RCP4.5 under "mitigation" scenario in the 2060s **(c, g)**, and

between G4 and RCP8.5 under "mitigation" scenario in the 2060s **(d, h)** based on ISIMIP **(a-d)** and WRF **(e-h)** results. Excessive collinearity variables have been removed (Fig. S18 shows the results without this procedure). Stippling indicates grid points where differences or changes are not significant at the 5% significant level according to the Wilcoxon signed rank test.

We firstly project the change of PM$_{2.5}$ under G4 and the aerosol mitigation scenario in 2060s relative to 2010s (Fig. 11a, e). Both ISIMIP and WRF project PM$_{2.5}$ decreases in most areas, especially in Tianjin and Langfang, but PM$_{2.5}$ decreases more under ISIMIP than WRF. PM$_{2.5}$ concentration decreases by 7.6 μg/m$^3$ over Beijing-Tianjin province in ISIMIP, and decrease by 5.4 μg/m$^3$ in WRF (Table S3). PM$_{2.5}$ concentration is 0.5-8 μg/m$^3$ higher in northern Beijing under G4 ("mitigation") than that during the 2010s in WRF. To show the impact of emission reductions, we compare the PM$_{2.5}$ concentration between aerosol "baseline" and "mitigation" scenarios under G4 in the 2060s (Fig. 11b, 11f), and compare the "mitigation" PM$_{2.5}$ concentration under G4 and the RCP scenarios in the 2060s to clarify the effect of geoengineering compared with climate warming. Compared with "baseline" scenario, PM$_{2.5}$ concentration is less under "mitigation" scenario as expected in both ISIMIP and WRF under G4 (Fig. 11b, 11f), and has a similar spatial pattern with that in Fig. 11a and 11e. Compared with RCP4.5 and RCP8.5, PM$_{2.5}$ concentration under G4 are higher over the Beijing-Tianjin province in ISIMIP results (Fig. 11c-11d), but with large differences between the 4 ESMs. G4 PM$_{2.5}$ is simulated greater than in RCP scenarios under HadGEM2-ES and BNU-ESM (Fig. S19k, l, o, p), but there are insignificant differences in most areas under the two MIROC models (Fig. S19c, d, g, h). PM$_{2.5}$ concentrations are larger between G4 and RCP8.5. WRF simulations shows similar changes in PM$_{2.5}$ between G4 and RCPs as ISIMIP over Beijing-Tianjin province (Fig. 11g-h).

### 3.3.2 PM$_{2.5}$ meteorological and emissions controls in the 2060s

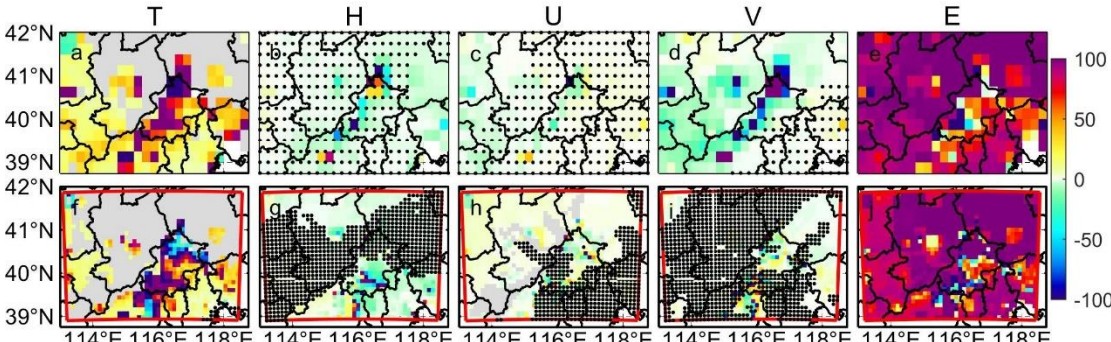

**Figure 12.** Contribution of climate factors (temperature/T, humidity/H, zonal wind/U, meridional wind/V) and emission (E) to changes in monthly PM$_{2.5}$ concentration ($\Delta$PM$_{2.5}$) in 2060s under G4 ("mitigation") relative to 2010s. Top figures **(a-e)** are ISIMIP results, and bottom figures **(f-j)** are WRF results. Stippling indicates the changes are insignificant at the 5% significant level in the Wilcoxon test. The grey areas represent the collinearity in the MLR, and they exist in the panel a, f

and h.

Next, we quantify the contribution of different meteorological factors and PM$_{2.5}$
emissions to ΔPM$_{2.5}$ between G4 ("mitigation") in the 2060s and the 2010s (Fig. 12).
Both ISIMIP and WRF results show that the increase of temperature and decrease of
PM$_{2.5}$ emission play positive roles in reducing PM$_{2.5}$ concentration. ISIMIP results (Fig.
12a-e), suggest that the projected increase of temperature could explain 0-20% of the
decrease of PM$_{2.5}$ concentration, and decrease of PM$_{2.5}$ emission could explain more
than 90% of changes in PM$_{2.5}$ concentration differences in most of areas. Changes in
humidity and westerly winds (positive U-wind) do not cause significant changes in
ΔPM$_{2.5}$, but projected increases southerly wind (positive V-wind) is detrimental to the
decrease in PM$_{2.5}$ concentration, and has a 0-10% negative effect on ΔPM$_{2.5}$ in
Zhangjiakou. WRF results show similar spatial pattern in effect of temperature and
emission on ΔPM$_{2.5}$ with ISIMIP results. Although temperature is projected to increase
over the whole domain (Fig. S22), there are negative contributions on ΔPM$_{2.5}$ to the
north of Beijing due to increase of PM$_{2.5}$ caused by the negative correlation between
PM$_{2.5}$ and its emissions (Fig. S26). The ~1-2% increase of humidity leads to ~10%
increase of PM$_{2.5}$ concentration in the south of Beijing (Fig. 12g), and 0.2-0.3 m/s
deceases of U-wind leads to 0-10% increase of PM$_{2.5}$ concentration in Zhangjiakou (Fig.
12h). The changes in each factor in ISIMIP and WRF results are shown in Fig. S21 and
Fig. S22, respectively.

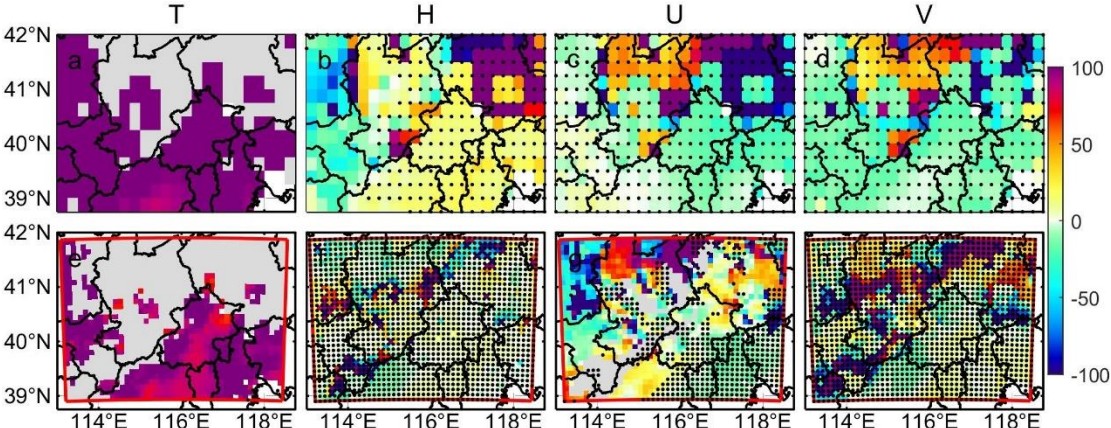

**Figure 13.** Contribution of climate factors (as in Fig. 12) to changes in monthly PM$_{2.5}$ concentration
in 2060s under G4 with aerosol "mitigation" relative to 2060s under RCP4.5 with aerosol
"mitigation". Top figures **(a-e)** are ISIMIP results, and bottom figures **(f-j)** are WRF results.
Stippling indicates the changes are insignificant at the 5% significant level in the Wilcoxon test. The
grey areas represent the collinearity in the MLR, and they exist in the panel a, f and h.

Now we explore the contribution of each meteorological factor to ΔPM$_{2.5}$ between G4
("mitigation") and RCP4.5 ("mitigation") in the 2060s (Fig. 13). The higher PM$_{2.5}$
under G4 is mainly caused by the lower temperature. In ISIMIP, lower temperature
explains more than 90% (100% in some places) of the raised PM$_{2.5}$ relative to RCP4.5,

although the increase of humidity is also helpful to lower $PM_{2.5}$ in the western domain (Fig. 13a-b). Humidity can increase suspended particle mass and coagulation, promoting deposition (Li et al., 2015). The contribution of differences in U-wind and V-wind on $\Delta PM_{2.5}$ is insignificant (Fig. 13c-d). In WRF, the projected lower temperatures explain more than 70% of the higher $PM_{2.5}$ under G4 relative to RCP4.5 (Fig. 13e). Although the increase of southerly (V) wind contributes 10-20% to the higher $PM_{2.5}$ in the northern domain under HadGEM2-ES and BNU-ESM (Fig. S24), it is insignificant in the ensemble (Fig. 13h). Decreased westerlies (U wind) explains about between +100% and -100% of $PM_{2.5}$ differences (Fig. 13g), since U-wind impacts vary spatially (Fig. S26).

### 3.3.3 $PM_{2.5}$ impact on health risks now and in the 2060s

Changes in RR of $PM_{2.5}$ for the 5 diseases under the geoengineering and global warming climate scenarios and different emission scenarios during 2060s relative to 2010s for the Beijing-Tianjin province are shown in Fig. 14. Present-day $PM_{2.5}$ related RRs are 1.32 (1.30), 1.37 (1.35), 1.46 (1.43), 1.83 (1.80) and 2.03 (1.99) for chronic obstructive pulmonary disease (COPD), ischemic heart disease (IHD), lung cancer (LC), lung respiratory infection (LRI) and stroke according to the ISIMIP (WRF) simulations (Fig. 14a). RR of LRI is the highest and COPD is the lowest in the five diseases, and WRF estimates of RR are 0.02-0.03 lower than those of ISIMIP. In both the "baseline" and "mitigation" emission scenarios, RRs will be lower under G4, RCP4.5 and RCP8.5 compared with the 2010s. Smaller RR reductions occur under G4 than under RCP4.5 and RCP8.5, and ISIMIP simulates larger reductions than WRF. This is because the $PM_{2.5}$ concentrations from ISIMIP are reduced more than with WRF (Table S3). Under the "baseline" emission scenario (Fig. 14b-d), the biggest reduction of RR for LRI is 0.047 under RCP8.5 in ISIMIP, and RRs for other diseases are projected to reduce by no more than 0.02. Under the "mitigation" emission scenario (Fig. 14e-g), reductions in RRs are 3-6 times greater.

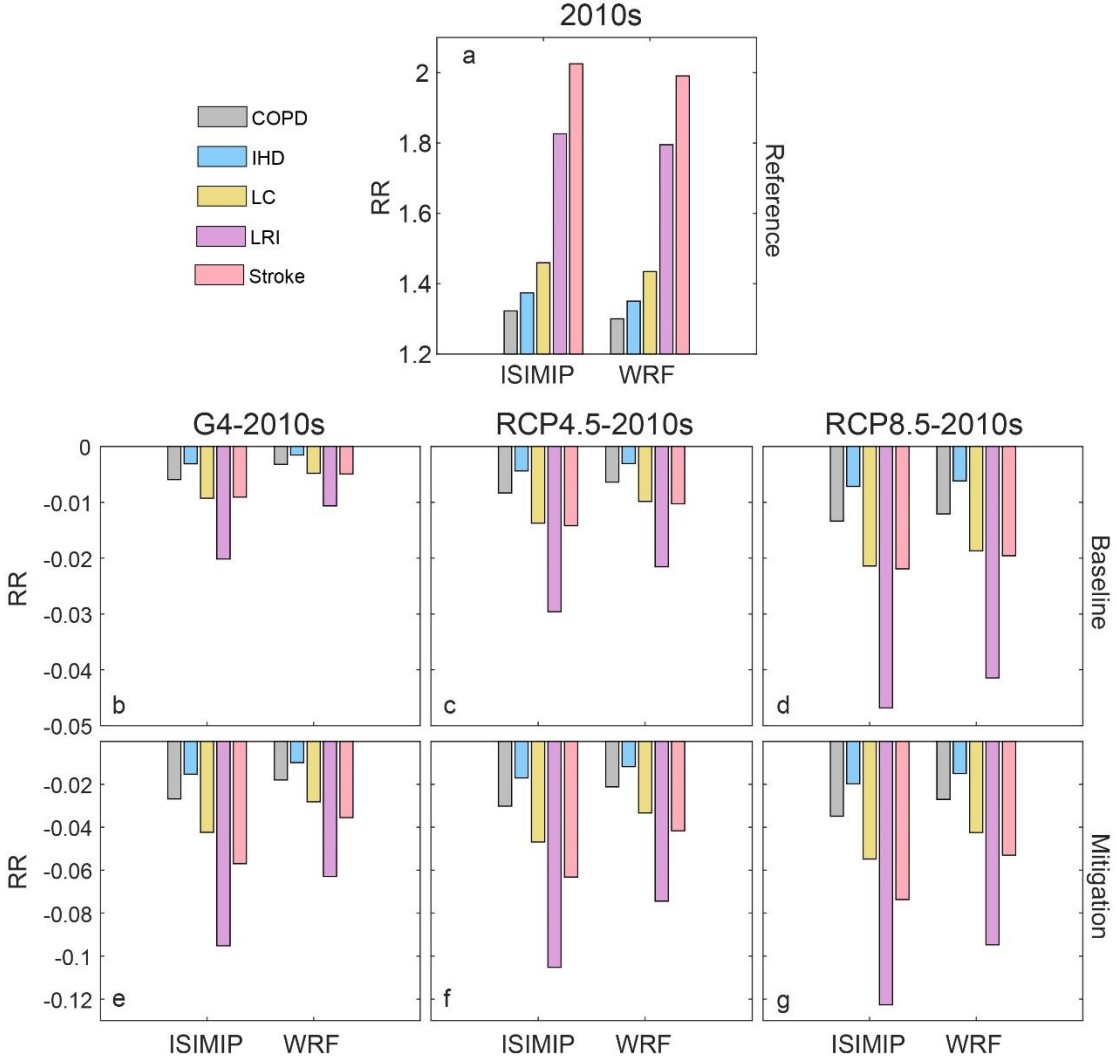

**Figure 14.** Average population-weighted relative risks of $PM_{2.5}$ related 5 diseases in 2010s **(a)** and its changes between G4 and 2010s **(b, e)**, between RCP4.5 and 2010s **(c, f)** and between RCP8.5 and 2010s **(d, g)** in Beijing-Tianjin province based on the ISIMIP and WRF results, respectively. $PM_{2.5}$ concentration is based on the "baseline" emissions under G4, RCP4,5 and RCP8.5 in the middle 3 figures **(b-d)**, and it is based on the "mitigation" emissions under G4, RCP4,5 and RCP8.5 in the bottom 3 figures **(e-g)**.

## 4. Discussion

## 4.1 Apparent temperature

Both ISIMIP and WRF can reproduce the observed (CN05.1) spatial patterns and seasonal variabilities of apparent temperature in the region around Beijing. WRF shows warm biases in AP during all months relative to CN05.1 due to warmer temperatures in urban areas, with the exception of BNU-ESM and HadGEM2-ES driven summers (Fig. S13). Both ISIMIP and WRF tend to overestimate population weighted NdAP_32 by

370% and 590%, respectively. These large discrepancies are due to relatively small overestimates of the likelihood of the tails of the probability distributions which leads to a dramatic increase in the frequency of extreme climate events (Dimri et al., 2018; Huang et al., 2021). AP is about 1.5°C warmer than 2 m temperature over the Beijing and Tianjin urban areas in summer due to higher vapor pressures amplifying warmer urban temperatures, and this is despite humidity being lower over the cities. Under high humidity conditions, a slight increase in temperature will cause a large increase in heat stress (Li et al., 2018; Luo and Lau, 2019). AP is nearly 4°C colder than 2 m temperature in winter due to wind speed (Fig. 2d). Differences between AP and 2 m temperature (AP-T) during summer are greater in urban areas than neighboring rural areas.

The apparent temperatures in Beijing Tianjin urban areas under G4 in the 2060s are simulated to be 1°C and 2.5°C lower than RCP4.5 and RCP8.5, although AP would be higher than in the recent past. The cooling effect of G4 relative to RCP4.5 and RCP8.5 is greatest under HadGEM2-ES (Fig. S14, S15), due to the ESM having largest temperature differences between scenarios (Wang et al., 2022). WRF downscaling produces reduced seasonality in AP compared with ISIMIP, and WRF produces relatively cooler summers and warmer winters than ISIMIP, and so much less differences in apparent temperature ranges (Fig. 15). Differences in AP between G4 and the RCP scenarios are mainly driven by temperature. In all scenarios and downscalings AP rises faster than the temperature due to decreased wind speeds in the future (Li et al., 2018; Zhu et al., 2021) but mainly because of rises in vapor pressure driven by rising temperatures. This effect occurs despite the general drying expected under solar geoengineering (Bala et al., 2008; Yu et al., 2015).

The NdAP_32 under G4 is projected to decrease by 8.6 days per year by ISIMIP and 6.3 days per year by WRF relative to RCP4.5 for Beijing-Tianjin Province. Much larger reductions in NdAP_32 of 33.5 days per year (ISIMIP) and 19.6 days per year (WRF) are projected relative to RCP8.5. Differences between scenarios in frequency of dangerously hot days are far larger using ISIMIP statistical downscaling than using WRF. This is another impact of the reduced seasonality of WRF compared with ISIMIP (Fig. 15).

The higher resolution WRF simulation produces a much larger range of apparent temperatures across the domain than CN05.1 and ISIMIP downscaling. This increased variability makes reaching a statistical significance threshold more challenging for WRF than ISIMIP results. Despite this, the ESM-driven differences in WRF output are less than from ISIMIP, reflecting the physically based processes in the dynamic WRF simulation. This reduces the impact of differences in ESM forcing at the domain boundaries with WRF compared with the statistical bias correction and downscaling methods. Although there are some uncertainties between models and downscaling methods, G4 SAI can not only reduce the mean apparent temperature but also decrease the probability of PDF tails (extreme events) in summer.

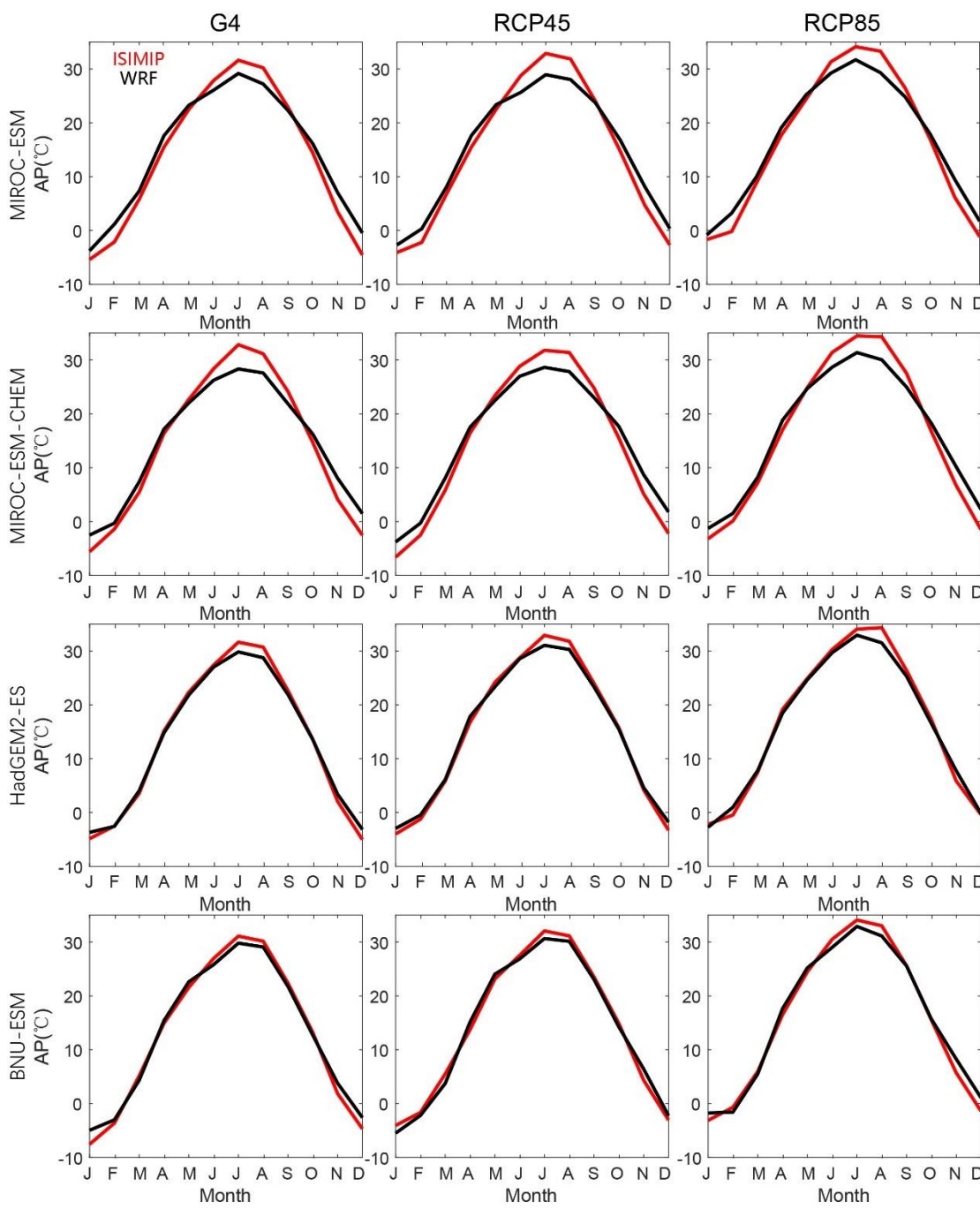

**Figure 15.** Seasonal cycles of apparent temperature from MIROC-ESM, MIROC-ESM-CHEM, HadGEM2-ES and BNU-ESM under G4, RCP4.5 and RCP8.5 in Beijing-Tianjin urban areas during 2060s based on ISIMIP (red) and WRF (black) methods.

## 4.2 PM$_{2.5}$

We established a set spatially gridded MLR models based on the 4 ESMs downscaled variables under ISIMIP and WRF. The meteorological factors impact PM$_{2.5}$ in complex ways, but the simple spatially gridded MLR models display enough skill to make some illustrative projections of future PM$_{2.5}$ explaining about 70% of the variance during the historical period. PM$_{2.5}$ concentration is correlated with emissions and anti-correlated

with temperature in most parts of the domain (Fig, S25-S26). Increased turbulence increases diffusion of $PM_{2.5}$ (Yang et al., 2016), and higher temperatures increase evaporation losses (Liu et al., 2015) of ammonium nitrate (Chuang et al., 2017), and other components (Wang et al., 2006). Humidity may have both positive and negative effects on $PM_{2.5}$ (Chen et al., 2020). It causes more water vapor to adhere to the surface of $PM_{2.5}$, thereby increasing its mass concentration and facilitating aerosol growth (Cheng et al., 2017; Liao et al., 2017). However, when the humidity exceeds a certain threshold, coagulation and particle mass increases rapidly, promoting deposition (Li et al., 2015). So, the slope coefficients between $PM_{2.5}$ and humidity are positive in low humidity areas, including southern plain and the Beijing-Tianjin province, but negative in some northern mountain areas (Fig. S25, S26).

There are large spatial differences in wind speed and direction impacts on $PM_{2.5}$. Yang et al. (2016) found that weaker northerly and westerly winds tend to increase the $PM_{2.5}$ concentration in northern and eastern China, respectively. The effects of wind direction depend on the distribution of emitted $PM_{2.5}$ and the condition of the underlying surface (Chen et al., 2020). Most sources of $PM_{2.5}$ lie to the south of our domain, relatively clean conditions prevail to the north, so northly winds tend to advect clean air, while southerlies bring high concentrations of aerosols. Weak winds tend to increase $PM_{2.5}$ and smog formation due to sinking air and weak diffusion (Su et al., 2017; Yang et al., 2017).

Xu et al. (2021) projected 2030 $PM_{2.5}$ concentrations will decrease by 8.8% and 5.5% under RCP4.5 and RCP8.5 respectively relative to 2015. Wang et al. (2021) also projected decreasing trends in China under RCP4.5 and RCP8.5 during 2030-2050. There were seasonal changes in $PM_{2.5}$ concentration differences between RCP4.5/8.5 scenarios and the historical scenario near the Bohai Sea (Dou et al., 2021). However, there are also some simulations where $PM_{2.5}$ concentrations increase in warmer climates. Hong et al. (2019) suggest that annual mean $PM_{2.5}$ concentrations will increase 1-8 $\mu g/m^3$ in an area including Beijing and Tianjin under RCP4.5 during 2046-2050, compared with 2006-2010. These inconsistent responses are mainly caused by the differences in the selection of ESMs, chemical transport models and climate/emission scenarios. Different RCP scenarios not only correspond to different future climate states, but also have different anthropogenic emissions of air pollutants. In our study, we do not consider the $PM_{2.5}$ emission differences between RCP4.5 and RCP8.5, and instead applied the ECLIPSE $PM_{2.5}$ emission scenarios in our MLR projection.

Emissions reductions are expected to play the dominant role in the decrease of $PM_{2.5}$ concentrations under G4 aerosol "mitigation" in 2060s (Fig. 12). Meteorological changes under the different future scenarios make much smaller changes as evidenced by the scenarios using "baseline" – that is present day $PM_{2.5}$ emissions, with decreases in mean annual concentration of 1.0 (1.3), 1.8 (2.0), 3.3 (3.2) $\mu g/m^3$ over Beijing-

Tianjin province under G4, RCP4.5 and RCP8.5 with WRF (ISIMIP), (Table S3), which
are mainly caused by the temperature increases (Fig. 13). The negative relationships
between emission and $PM_{2.5}$ concentration result in the increase of $PM_{2.5}$ under G4
("mitigation") relative to 2010s in the north of Beijing with WRF. This may be due to
changes in $PM_{2.5}$ out of the domain being opposite to those in domain during the MLR
fitting period, since relocation of polluting sources from the urban areas mainly to the
west, was occurring over the calibration period. The accuracy of $PM_{2.5}$ emission data is
also crucial for training MLR models, and $PM_{2.5}$ data was sparse before 2013, relying
on reconstructions based on satellite optical depth estimates. Although both increase of
temperature and decrease of emission explain more than 90% of the decrease in $PM_{2.5}$
in most areas, there are large spatial differences due to wind and humidity. On the one
hand, there is uncertainty in the differences in changes of wind speed and humidity
between different ESMs and downscaling methods; on the other hand, the complex
physical relationship between them and $PM_{2.5}$ also increases uncertainties. Reductions
in $PM_{2.5}$ in the future are projected to decrease $PM_{2.5}$ related health issues, although its
effect on different diseases are different. Changes in $PM_{2.5}$ related risk between G4 and
RCPs are from 1-3%, with $PM_{2.5}$ emissions policy dominating differences over climate
scenario.
There are some differences in projecting $PM_{2.5}$ concentration between WRF and ISIMIP
methods. Compared to the 2010s reference, $PM_{2.5}$ concentration in ISIMIP are
projected to decrease more than using WRF in G4 under the "mitigation" scenario
during the 2060s over the Tianjin province (Fig. 11a, e). However, the spatial patterns
of changes in $PM_{2.5}$ concentration between G4 and RCP4.5/8.5 under the "mitigation"
scenario during 2060s are similar (Fig. 11c-d, g-h). This means that the effects of
different downscaled methods on projecting $PM_{2.5}$ are small if we only consider the
climate change alone without considering emissions changes. Due to the larger
regression coefficient of emissions in the MLR under the ISIMIP method (Fig. S25,
S26), the negative changes in $PM_{2.5}$ concentration are larger between "mitigation" and
baseline under G4 during 2060s than that under the WRF method. Correspondingly, the
ISIMIP method has a greater reduction in $PM_{2.5}$ related RR than WRF under three future
climate scenarios during the 2060s.
Eastham et al. (2018) deduced from experiments using 1 Tg/yr SAI in a coupled
chemistry-transport model directly simulating atmospheric chemistry, transport,
radiative transfer of UV, emissions, and loss processes, that per unit mass emitted,
surface-level emissions of sulfate result in 25 times greater population exposure to
$PM_{2.5}$ than emitting the same aerosol into the stratosphere. The G4 experiment specifies
5 Tg/yr injection rate, which over our domain would equate to 1450 t/yr if it was
deposited uniformly globally (which it certainly would not be). Reducing this by the
1/25 factor amounts to 58 t/yr which can be compared with present $PM_{2.5}$ emissions of
around $3.3 \times 10^5$ t/year in our domain. If we consider the aerosol deposition under G4
scenarios, $PM_{2.5}$ concentration will be 0-1 $\mu g/m^3$ higher than that without due to
deposition of the SAI aerosols (Fig. S27), and RR is projected to increase by 0.01% for
Beijing-Tianjin province (Table S4). This comparison suggests that tropospheric
emissions will be much more important for human health in our domain than from the
SAI specified by G4.

The most important change in $PM_{2.5}$ will come from emissions reductions, with the
different weather conditions under both G4 and RCP scenarios making relatively little
practical differences in concentrations. $PM_{2.5}$ concentration is expected to decrease
significantly (ISIMIP: -7.6μg/m$^3$, WRF: -5.4 μg/m$^3$) in the Beijing-Tianjin province,
but they will still not meet either Chinese or international standards. The temperature
under G4 is lower than that under RCP4.5 and RCP8.5 scenarios, which makes the
$PM_{2.5}$ concentration under G4 higher. But the difference in $PM_{2.5}$ between the two is
small and even within uncertainty due to projected differences in humidity and wind.
Potentially improved estimates from more complex models such as WRF-Chem,
CMAQ and GEOS-Chem over the simple MLR methods used here will be of limited
value unless the differences between the ESM driving these models is reduced. It can
be confirmed that emission policies based on the 13$^{th}$ Five Year Plan are not enough,
and higher emission standards need to be developed for a healthy living environment.

Our study did not consider the impacts of socio-economic pathways on $PM_{2.5}$ future
emissions, instead we explore the meteorological differences between the SAI G4
scenario and the greenhouse gas RCP4.5/RCP8.5 on $PM_{2.5}$ concentrations. $PM_{2.5}$
emissions were defined by the uncontrolled ("baseline") and a scenario where
technological intervention ("mitigation") reduces emissions. There are some limitations
in our study. Firstly, the HTAP_V3 dataset only includes anthropogenic $PM_{2.5}$ emission,
not natural $PM_{2.5}$ emission. Natural $PM_{2.5}$ will also change in the future under changing
climate. The sources of natural $PM_{2.5}$ include the sandstorms that sometimes occur in
spring as extreme winds mobilize dry unvegetated soils. These relatively extreme
conditions are difficult to simulate in ESM and subject to land use policy e.g., the
numerous ecosystem service measures undertaken by China over the last five decades
(Miao et al.,2015). Secondly, although $PM_{2.5}$ concentration includes both primary and
secondary $PM_{2.5}$ during model training, we do not consider the precursor gases for
secondary $PM_{2.5}$ directly. The sensitivity of MLR may diminish at the high $PM_{2.5}$ values
when secondary $PM_{2.5}$ dominates the variability of total $PM_{2.5}$ (Upadhyay et al., 2018).
Thirdly, we only consider the effect of dominant near-surface meteorological variables
on the $PM_{2.5}$. However, the vertical transport of pollutants related to vertical
atmospheric stability should not be ignored (Lo et al., 2006; Wu et al., 2005), and this
may contribute to the differences in RCP4.5 scenario from our MLR model and more
sophisticated simulations (Fig. S7). Finally, although it is insignificant for the Beijing
and Tianjin provinces, the MLR model suffers collinearity problems in some areas.
These factors play smaller roles as we are mainly considering changes in $PM_{2.5}$
concentration between different climate scenarios. Nevertheless, projection for changes

in $PM_{2.5}$ between SAI scenarios and per greenhouse gas scenarios would be valuable for global air quality impacts from geoengineering.

# 5. Conclusion

Our study on thermal comfort and aerosol pollution under geoengineering scenarios for the Beijing megalopolis may be useful across the developing world, which is expected to suffer disproportionate climate impact damages relative the global mean, while also undergoing rapid urbanization. Assessing health impacts and mortality due to heat stress and $PM_{2.5}$ under greenhouse gas scenarios should consider urbanization and the change to concrete surfaces from vegetation that leads to differences in heat capacities, rates of evapotranspiration, and hence humidity and apparent temperature. These require downscaled analyses, accurate meteorological and high-resolution land surface datasets, and industrial development scenarios.

In our analysis we assumed the urban area did not change over time, and also that population remains distributed as in the recent past. This may be reasonable in the highly developed and relatively mature greater Beijing-Tianjin region but should be considered in rapidly urbanizing regions elsewhere. There certainly will be changes over time in the radiative cooling from surface pollution sources. $PM_{2.5}$ is a health issue in many developing regions (Ran et al., 2023), but as wealth increases efforts to curb air pollution generally clean the air. This has clear health benefits, but also removes aerosols from the troposphere that cool the surface. The urban areas that have higher apparent temperatures at present are also the areas with greatest aerosol load and hence greatest cooling. Once that is removed direct radiation, air temperatures and apparent temperatures will all rise – by several degrees (Wang et al., 2016). So, a future more comprehensive health impact study would include both the negative health impacts of aerosol pollution and the potential cooling effects those aerosols produce. Additionally, the formulation of apparent temperature used does not consider the effect of radiation on human comfort (Kong and Huber, 2022). When $PM_{2.5}$ levels are high there is no shade because the sky is milky-white, similarly SAI will brighten the sky (Kravitz et al., 2012). Comfort is increased in clear sky conditions when shade is readily found.

The changes simulated to relative risk from increased $PM_{2.5}$ under the G4 SAI scenario are about 1-3% worse than under RCP4.5, mainly because of lower temperatures under G4. The difference this would make to the overall health burden under SAI depends on the range of other impacts that include changes in apparent temperature we discuss. G4 reduces the number of days with AP>32 (when extreme caution is advised) by 6-8 per year relative to RCP4.5 and by 20-34 relative to RCP8.5. But G4 itself will still increase these extreme caution days by 1-20 relative to conditions in the 2010s. Lowering PM2.5 emissions will increase ground temperatures and the associated risk of dangerous apparent temperatures will also increase rapidly as the distribution of temperatures is shifted making presently rare hot events into much more frequent heat waves.

**Code and data availability**

All ESM data used in this work are available from the Earth System Grid Federation (WCRP, 2021; https://esgf-node.llnl.gov/projects/cmip6, last access: 14 July 2021). The WRF and ISIMIP bias-corrected and downscaled results are available for the authors on request. WRF and ISIMIP codes are freely available at the references cited in the methods sections.

**Supplement link**

The link to the supplement will be included by Copernicus.

**Author contribution**

JCM and LZ designed the experiments, JW performed the simulations. All the authors wrote the manuscript.

**Competing interests**

The authors declare that they have no conflict of interest.

**Disclaimer**

Publisher's note: Copernicus Publications remains neutral with regard to jurisdictional claims in published maps and institutional affiliations.

**Special issue statement:**

This article is part of the special issue "Resolving uncertainties in solar geoengineering through multi-model and large-ensemble simulations (ACP/ESD inter-journal SI)". It is not associated with a conference.

**Acknowledgements**

We thank the editor and two constrictive referees for improving the manuscript. This work relies on the climate modeling groups participating in the Geoengineering Model Intercomparison Project and their model development teams; the CLIVAR/WCRP Working Group on Coupled Modeling for endorsing the GeoMIP; and the scientists managing the earth system grid data nodes who have assisted with making GeoMIP output available. This research was funded by the National Key Science Program for Global Change Research (2015CB953602).

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
