# Peer review of "Changes in apparent temperature around the"

_Earth System Dynamics, 2022_

## Author Comment (AC1)

Referee's comments are in red, our reply in black, quotes in the revised manuscript in blue.

**Referee 1's comments**

**General Comments**

The authors seek to address the question of how much apparent temperature in Beijing will vary under different future scenarios of climate change (including geoengineering). This includes an analysis of whether different downscaling methods – either statistical or dynamical – yield different results. They find that, although both methods (when applied to results from 4 global ESMs) yield roughly similar results for the present day, the same is not true when inspecting the effects that climate change and geoengineering will have. The study highlights the important issue of changes in human-relevant variables such as apparent temperature (and the number of times a threshold is crossed) rather than relatively abstract variables such as global mean surface temperature.

I struggled with this review because I could not clearly identify the core contribution. The base idea of whether statistical downscaling or dynamical scaling results in different outcome estimates is certainly important, but this question has been thoroughly discussed in a companion paper by the authors which looks at the same data for the same domain from the same models, and was submitted recently to this journal (https://esd.copernicus.org/preprints/esd-2022-35). The remaining question is whether apparent temperature is differently affected than more conventional meteorological variables, which is a relatively boutique concern. The methods used to address this questions are nonetheless appropriate, and the data produced generally support the conclusions. However, the existence of the companion paper (which I recognize the authors do cite) makes the contribution of this manuscript incremental.

The use of multiple downscaling techniques with multiple models is interesting and well executed, and it is particularly encouraging to see applications to health-relevant outcomes. The biggest issue is a lack of significant impact, although I also have some methodological concerns. I have laid these out in detail below, starting with major comments. If the paper can be focused more heavily on outcomes – in particular, the effect that downscaling has on health-relevant impacts – then I believe it could significantly improve its relevance and impact. This would also help to address the issue that the paper is not particularly interdisciplinary, which is a stated requirement of ESD. As such, in its current state I cannot recommend it for publication.

Reply: We thank your constructive comments, which help us clarify and improve the study vastly. There are two main problems, one is the lack of innovation, the other is the method of processing data. These two issues are also detailed in the major comments. We have responded to the major comments below one by one.

**Major comments**

The greatest issue is the lack of a clear and impactful outcome. The methods applied are interesting in large part because they look at interesting scenarios (RCPs versus geoengineering versus recent past) and include a significant problem (the performance of statistical versus dynamical downscaling). However, these issues are the focus of a paper which is already under review, and as such cannot be the major novelty of a second manuscript. I therefore assume that the major conclusions regard the question of change in apparent temperature, with the authors finding that changes in apparent temperature will be greater under RCP 8.5 than under a geoengineering scenario, and that this is mostly because of increases in temperature. The issues I perceive here are twofold. Firstly, apparent temperature – while an important metric – is just one metric of impact, and a relatively straightforward one which is (evidently) mostly just reflecting changes in temperature. The manuscript would be greatly improved if multiple outcomes were assessed rather than just one, to see if the different downscaling methods have different impacts on such outcomes. This could include, for example, regional air pollution (if reported in any of the ESMs). Alternatively, a deeper analysis of the likely consequences – for example by attempting to quantify the differences in health outcomes or costs, and the degree to which different demographics or sub-populations are affected – would help to improve the interdisciplinarity of the manuscript. Secondly, the current analysis is somewhat limited, being mostly observational (report differences) rather than explanatory. The manuscript would be greatly improved if the authors could provide mechanistic explanations for their findings; why, for example, does WRF-based downscaling seem to result in such a different seasonality in AP – T compared to statistical downscaling?

Reply: Thanks for these thoughts. We wanted to address a problem related to impacts of changes in the fundamental weather fields. The suggestion of looking at regional pollution is most relevant we felt. However, essentially the output from ESM on $PM_{2.5}$ simulations is not good (e.g. Ran et al., 2022). So we explored other ways of projecting air pollution. Based on our existing downscaling data, we further explored the impact of geoengineering on $PM_{2.5}$ in the Beijing-Tianjin region using the multiple linear regression model. We also explore the changes in $PM_{2.5}$ related relative risks of 5 main diseases.

References

Ran, Q., Lee, S., Zheng, D., Chen, S., Yang, S., Moore, J., and Dong, W.: Potential Health and Economic Impacts of Shifting Manufacturing from China to Indonesia or India, Sci. Total Environ., 855, 158634, https://doi.org/10.1016/j.scitotenv.2022.158634,2023.

The details are as follows:

Introduction

[revised manuscript text omitted]

Supplement

[Figure]

**Figure S1.** Annual mean PM$_{2.5}$ concentration (a, b) and PM$_{2.5}$ emissions (c, d) map for Beijing and surrounding areas during 2008 (a, c) and 2017 (b, d).

[Figure]

**Figure S2.** Annual PM$_{2.5}$ emissions from different sources in Beijing under the ECLIPSE V6b baseline scenario *(Source: GAINS East Asia online (iiasa.ac.at))*.

[Figure]

**Figure S3.** Distribution of observed PM$_{2.5}$ concentration (ug/m$^3$) from ChinaHighPM$_{2.5}$ **(a)** and estimated ensemble-mean PM$_{2.5}$ concentration from MLR under ISIMIP **(b)** and WRF **(c)** results for Beijing and surrounding areas during 2008-2017.

**Table S1.** Difference of PM$_{2.5}$ concentration between different scenarios for the Beijing-Tianjin province as defined in Fig. 1b during 2060-2069. The PM$_{2.5}$ emission scenarios used in each climate scenarios are in parentheses. Bold indicates the differences or changes are significant at the 5% significant level according to the Wilcoxon signed rank test. (Units: μg/m$^3$)

| Model | G4 (mitigation) -2010s (reference) | | G4 (mitigation) -G4 (baseline) | | G4 (mitigation) -RCP4.5(mitigation) | | G4 (mitigation) -RCP8.5(mitigation) | |
|---|---|---|---|---|---|---|---|---|
| | WRF | ISIMIP | WRF | ISIMIP | WRF | ISIMIP | WRF | ISIMIP |
| MIROC-ESM | **-4.5** | **-6.3** | **-3.1** | **-3.8** | 0.5 | 0.7 | **2.3** | **2.3** |
| MIROC-ESM-CHEM | **-6.0** | **-7.4** | **-4.9** | **-5.3** | 0.5 | -0.2 | **1.9** | 0.6 |
| HadGEM2-ES | **-4.8** | **-6.8** | **-3.8** | **-6.8** | **1.4** | **1.3** | **2.6** | **2.6** |
| BNU-ESM | **-2.5** | **-5.5** | **-1.4** | **-5.0** | **0.8** | **1.1** | **2.4** | **2.2** |
| Ensemble | **-4.3** | **-6.5** | **-3.3** | **-5.2** | **0.8** | **0.7** | **2.3** | **1.9** |

[Figure]

**Figure S4.** Spatial patterns of PM$_{2.5}$ concentration difference (μg/m$^3$) between "mitigation" in the 2060s under G4 and 2010s **(a, e, i, m)**, between "mitigation" and "baseline" under G4 **(b, f, j, n)**, between G4 and RCP4.5 under "mitigation" scenario **(c, g, k, o)**, and between G4 and RCP8.5 under "mitigation" scenario **(d, h, l ,p)** based on ISIMIP results. From top to bottom are MIROC-ESM **(a-d)**, MIROC-ESM-CHEM **(e-h)**, HadGEM2-ES **(i-l)** and BNU-ESM **(m-p)** respectively. Stippling indicates grid points where differences or changes are not significant at the 5% significant level according to the Wilcoxon signed rank test.

[Figure]

**Figure S5.** Same as Fig. S4, but by WRF.

[Figure]

**Figure S6.** Spatial pattern of changes in temperature (T/°C), humidity (H/%), zonal wind (U/m s$^{-1}$), meridional wind (V/m s$^{-1}$) and PM$_{2.5}$ emissions (E/kg m$^{-2}$ s$^{-1}$) under G4 ("mitigation") in the 2060s relative to 2010s in ISIMIP. Stippling indicates grid points where differences or changes are not significant at the 5% significant level according to the Wilcoxon signed rank test.

[Figure]

**Figure S7.** Same as Fig. S6, but by WRF.

[Figure]

**Figure S8.** Spatial pattern of changes in temperature (T/°C), humidity (H/%), zonal wind (U/m s$^{-1}$) and

meridional wind (V/m s$^{-1}$) under G4 ("mitigation") relative to RCP4.5 ("mitigation") in the 2060s in ISIMIP. Stippling indicates grid points where differences or changes are not significant at the 5% significant level according to the Wilcoxon signed rank test.

[Figure]

**Figure S9**. Same as Fig. S8, but for WRF results.

[Figure]

**Figure S10**. Slope coefficients of MLR of temperature, humidity, u-wind, v-wind and emission for ISIMIP results during training period.

[Figure]

**Figure S11.** Similar as Fig. S10, but for WRF results.

[Figure]

**Figure S12.** Spatial pattern of changes in $PM_{2.5}$ ($\mu g/m^3$) between G4 with and without considering aerosol deposition due to SAI specified by G4.

**Table S2.** RRs of the 5 mortality endpoints under G4 with and without considering aerosol deposition from the G4 SAI specification in both PM$_{2.5}$ aerosol "baseline" and "mitigation" scenarios.

[revised manuscript text omitted]

I also have a methodological concern regarding the method used to try and separate out the roles of different meteorological variables in changes in AP. It is not clear to me why a linear regression is used. The expression for AP is a simple (albeit non-linear) combination of variables, which can be easily and explicitly broken down to find how each one contributes to changes in AP. I suggest the authors at least evaluate how their contributions change if they calculate them based on the degree to which excluding a factor changes AP (i.e. contribution of T to AP is estimated by calculating change in AP with no change in T, but including other factors). The authors could also consider defining the derivatives of AP with respect to each factor, given that these should be well defined (and include the Clausius-Clapeyron relationship directly).

Reply: The reason we used the regression approach is that this produces a least squares estimate of contributions. This is useful in many statistical applications and has desirable mathematical properties compared with, for example, absolute differences. However, it may not be the best choice here as the assumptions of Normality and homoscedasticity in the analysis are probably not true. The referee's suggestions are more localized estimators based around the mean values, which could be regarded as more statistically more robust, as they give less weight to outliers compared with minimizing squared anomalies. But these are reasonable alternatives and the gradient or Jacobian approach plays a role in statistical analyses. We compared the results using the regression method and the first method the referee suggests using. The detailed steps and results are as follows.

To determine the contribution of change in AP ($\Delta AP$) for each meteorological factor under different scenarios, we first calculate the $\Delta AP$ caused by individual changes in three factors as follows:

$$\Delta AP(X_i) = \begin{cases} f(T_b, H_a, W_a) - f(T_a, H_a, W_a), i = 1 \\ f(T_a, H_b, W_a) - f(T_a, H_a, W_a), i = 2 \\ f(T_a, H_a, W_b) - f(T_a, H_a, W_a), i = 3 \end{cases} \qquad (1)$$

Where daily $\Delta AP(X_i)$ are the $\Delta AP$ caused by individual changes in three factors: temperature $(X_1)$, humidity $(X_2)$ and wind speed $(X_3)$. $f()$ is the function of calculating AP. T, H, W are the daily temperature, humidity and wind speed respectively, and the subscripts $a$ and $b$ represent two different climate scenarios, respectively.

Then the contribution of each factor can be expressed as the ratio of the $\Delta AP$ caused by one factor alone to the total $\Delta AP$ caused by three factors.

$$C(X_i) = \frac{\overline{\Delta AP}(X_i)}{\sum_{i=1}^{3}\overline{\Delta AP}(X_i)} \qquad (2)$$

Where $C(X_{i(i=1,2,3)})$ is the contributions from changes in each factor to the $\Delta AP$, and $\overline{\Delta AP}(X_i)$ are the mean $\Delta AP(X_i)$. One thing to note is that due to the nonlinear relationship between factors, the total $\Delta AP$ caused by three factors is not strictly equal to the $\Delta AP$ itself.

We next replotted the figure 6 so we compare it with our previous plot.

[Figure]

**Alternative Figure 6.** The alternative method of calculating the contributions to seasonal changes of AP ($\Delta AP$) and the seasonal contribution of climatic factors to $\Delta AP$ for Beijing and Tianjin urban areas under ISIMIP and WRF between G4 and 2010s **(a)**, G4 and 2010s **(b)**, G4 and RCP4.5 **(c)** and G4 and RCP8.5 **(d)** based on ensemble mean results. Bold italic numbers and "*" above the columns indicate differences are significant at the 95% under the Wilcoxon test.

[Figure]

**Preferred original Figure 6.** The seasonal changes of AP (ΔAP) and the seasonal contribution of climatic factors to ΔAP for Beijing and Tianjin urban areas under ISIMIP and WRF between G4 and 2010s (**a**), G4 and 2010s (**b**), G4 and RCP4.5 (**c**) and G4 and RCP8.5 (**d**) based on ensemble mean results. Colors and numbers in each cell correspond to color bar, and "*" above the columns and in the cells indicate differences are significant at the 95% under the Wilcoxon test.

We calculated the differences of contribution (%) of each meteorology on changes in apparent temperature between alternative suggested method (**Alternative Figure 6**) and preferred original method (**Preferred original Figure 6**), as shown in the figure below.

[Figure]

**Figure.** The differences of contribution (%) of each meteorology on changes in apparent temperature between alternative suggested method and preferred original method.

The contributions of temperature and humidity are different using different methods, but the contribution of wind speed shows little change due to the linear relationship between wind speed and AP under either method. Changes in contribution from humidity is significant. In the referee's suggested method, the contribution of humidity

is influenced by the hybrid effect of temperature, with big changes under higher temperature in JJA. As we all know, AP will change a lot under high temperature, although humidity changes little. In panel a and b, the contribution of humidity under referee's method is higher than that under previous method, but the opposite in the panel c and d. This is because different reference scenarios have different effects when calculating the contribution of humidity. For example, when we calculate the contribution of humidity on AP between G4 and RCP4.5, we can get the value of contribution A (we maintain the temperature in the G4 scenario and the humidity changes with the scenario) and B (we maintain the temperature in the RCP4.5 scenario and the humidity changes with the scenario), but A is not equal to B.

In summary, if we use the suggested method, the sum of changes in AP caused by three factors is not strictly equal to the absolute change in AP and the contribution of humidity and temperature is different when we select different reference between two scenarios. Actually, there is no best way to calculate contributions. Of course there are uncertainties between different methods. We prefer our original method, so we retain it unchanged in our paper.

We changed the sentence in line 178 using the followed sentences

We use an MLR approach, since this minimizes the square differences from the mean across the dataset, with the attendant assumption of independence between the data. Alternatives may also be considered that e.g. minimize the impact of outliers by considering the magnitude of the differences, but we prefer to keep the attractive properties of a least squares approach.

Finally, much of the analysis is rather subjective (e.g. lines 253-257 – "little difference", "slightly worse", "slightly better"). I would recommend that the authors revise the text to make use of quantitative statements, in particular from line 219 onwards. Furthermore, statements such as "There are no models with obvious regional differences" (line 287), "AP changes … are essentially the same" (line 296), "all ESM reproduce the ERA5 pdf well" (line 261), "striking differences" (line 318) and "ERA5… probably does not account for the broad overestimate" (line 234) lack rigor and are difficult to interpret or verify without some context (what counts as a broad overestimate, or an obvious difference? How big would a difference in the change in AP have to be to not count as essentially the same? Why?). A particularly significant example is on line 255, where it is stated that BNU-ESM's performance is "slightly worse" than the other three models when using the ISIMIP method to inspect the recent past. This seems like a significant understatement; BNU-ESM's performance appears to be significantly worse than the other three models (r ~0.85 compared to ~0.92 for the other three), predicting both too many extreme low temperatures and not enough moderately low temperatures (see Figure 4). This is central to the manuscript, since WRF appears to be able to "save" BNU-ESM, bringing its performance to at least be similar to that of HadGEM (albeit still worse than MIROC-ESM[-CHEM]).

Reply: Thanks for your suggestions. we edited them.

Lines 219-241 are edited as follows. It is the same with descriptions in our reply to your second major comments.

[revised manuscript text omitted]

Minor comments

L45-47: Need citations to support idea that apparent temperature is actually an important variable

Reply: Done. I added the references.

Apparent temperature (AP), that is how the temperature feels, is formulated to reflect human thermal comfort and is probably a more important indication of health than daily maximum or minimum temperatures (Fischer et al., 2013; Matthews et al., 2017; Wang et al., 2021).

L164-166: The rationale for using NdAP_32 does not make sense to me. Since you are looking to identify an increase in the frequency of a rare event, why does the fact that it is rare mean that you should not use it? Similarly, why presume that the same outcome will apply for higher thresholds? I suggest revising the rationale.

Reply: We cannot simply use any threshold because the less frequent the threshold the more statistically uncertain is the estimate of its probability. For example the well-known estimate for the uncertainty in an estimate of uncertainty, s, is s/√(2n-2). So if we have only a very small number of instances of s, (that is n) then its uncertainty is very high. So, we must compromise in having a measure of extreme that represents the tail of the distribution, while at the same time being common enough for a reasonable sampling of its likelihood in the 50 years or so of simulations available. This is why we choose NdAP_32 rather than say NdAP_27 or NdAP_36.

In regard of the second point - we do not necessarily think that rarer events will be changed by the same amount as NdAP_32, in fact, often extremes change more than central parts of the distribution. This is a consequence the "fat tails" seen in most realworld climate distributions. The reason for the fat-tailed nature of real-world climate simulations probably relates to the long term spatial and temporal persistence (that is not simply autocorrelation) of processes rather than them being independent, and also to the presence of hysteresis behaviour (tipping points) in the system as it pushed further from the long term mean – for example in the fundamentally different physics at play on either side of the ice/water phase change. The fat tails implications for risk were examined in regard of economics by Weitzman's (2009) Dismal theorem which showed that since the likelihood of extreme fat tail probability distributions decay polynomially, the damage associated with them rises exponentially, thus leading to no bound when integrated to infinity. However, we do not think this is useful discussion in the manuscript. The issue is that we do not have the statistical power to discuss rarer extremes than NdAP_32 with the data available. This is what we tried to explain in the text in a simple way.

L168: "Since health impacts are more important where there are more people": this seems like a value judgement, and not (I think) the intended meaning. I recommended simply stating that you calculated population-weighted changes.

Reply: No it is not. There are no value judgements here at all. The value of human life is exactly the same in the sentence, i.e. each life is the same. There are simply more lives in urban areas than rural ones. Hence the phrase "more important". This is the same logic and values that suggest we should not be worried at all about human health impacts of climate change on Mars because there are no people there to be impacted.

The dark colors in Figure 6 make it nearly impossible to read the data.

Reply: We have changed the color bar in Figure 6.

[Figure]

**Figure 6.** The seasonal changes of AP (ΔAP) and the seasonal contribution of climatic factors to ΔAP for Beijing and Tianjin urban areas under ISIMIP and WRF between G4 and 2010s **(a)**, G4 and 2010s **(b)**, G4 and RCP4.5 **(c)** and G4 and RCP8.5 **(d)** based on ensemble mean results. Colors and numbers in each cell correspond to color bar, and "*" above the columns and in the cells indicate differences are significant at the 5% significant level under the Wilcoxon test.

Figure 7: please label the months.

Reply: Done.

[Figure]

**Figure 10.** Seasonal cycles of apparent temperature from MIROC-ESM, MIROC-ESM-CHEM, HadGEM2-ES and BNU-ESM under G4, RCP4.5 and RCP8.5 in Beijing-Tianjin urban areas during 2060s based on ISIMIP (red) and WRF (black) methods.

Throughout: it would be helpful to see the baseline (undownscaled) results alongside the downscaled results, so that the readers might know how significant the differences between ISIMIP and WRF are compared to the differences between the original and downscaled outputs.

Reply: We plot the ESMs original AP in Fig. 3. We added two sentences after line 222 and 229.

[Figure]

**Figure 3.** Top row: the spatial distribution of mean apparent temperature from CN05.1 **(a),** raw ESMs ensemble mean after bilinear interpolation **(b)**, 4-model ensemble mean after ISIMIP **(c)** and ensemble mean after WRF **(d)** during 2008-2017. Bottom row: the spatial distribution of annual mean number of days with AP > 32°C from CN05.1 **(e)**, ESMs **(f)**, ISIMIP **(g)** and WRF **(h)** during 2008-2017. Fig. S1 and Fig. S2 show the pattern of AP and NdAP_32 for the individual ESM.

While the raw AP from ESMs is overestimated in the Zhangjiakou high mountains and underestimated in the southern plain, and shares a similar pattern with temperature from ESMs (Wang et al., 2022). The raw ESM outputs were improved after dynamical and statistical downscaling.

ESMs tend to overestimate the number of days with AP>32°C in southeastern Beijing and for the whole Tianjin province.

---

## Author Comment (AC2)

Referee's comments are in red, our reply in black, quotes in the revised manuscript in blue.

**Referee 2's comments**

**General Comments**

The purpose of this work was to compare the changes in apparent temperature across three future scenarios using two different downscaling techniques. The authors find that although both downscaling methods using ISIMIP and WRF reproduce historical observations, projections to future scenarios produce differing results. In general the authors conclude that changing temperature contributes most to changes in apparent temperature which is driven by a combination of 2-m temperature, relative humidity and windspeed to more accurately capture the physiological impact of a warming climate. They find that the occurrence of days exceeding a 32 deg C apparent temperature threshold across the Beijing-Tianjin megalopolis will increase in frequency under RCP 4.5 and RCP 8.5. They draw attention to GeoMIP scenario G4, designed to test SAI for its ability to mitigate risks from an RCP 4.5 scenario, finding that individual ESMs show no statistically significant differences in the number of days exceeding 32 deg C apparent temperature between G4 and RCP 4.5.

This paper is companion to a manuscript which focuses on the impacts of using different downscaling methods in the Beijing-Tianjin region. Therefore, my interpretation is that the core contribution of this paper is to quantify the effect of proposed SAI for this region, and to compare this effect across two downscaling methods. However, in some ways this comparison seemed like an afterthought, and the paper discussion centered on the differences in apparent temperature across the two downscaling methods. Therefore, I would agree with reviewer 1 in claiming that this paper is providing only an incremental contribution with this manuscript. If the authors were to reframe this piece to focus on key differences in SAI forcing vs the RCP 4.5 scenario using downscaling to identify sources of uncertainty in response, then I believe this would provide novel insight into SAI as a proposed technique.

Reply: Thanks for your constructive suggestions. As noted we mainly explore the impact of geoengineering on apparent temperature, which is affected by the combined effects of temperature, humidity and wind speed. It is clear that the change in temperature dominates the change in apparent temperature. To study more impacts of changing meteorological conditions under SAI on human health, we develop an analysis of air pollution, using $PM_{2.5}$ since this is the dominant component at present in the region in our revised manuscript. The specific revisions are shown in response to referee 1.

**Major Comments**

Use of the WRF and ISIMIP downscaling techniques across the four ESMs used was technically interesting and well executed. The use of apparent temperature was also useful, but given the results were largely driven merely by changes in 2 m surface temperature, it left the reader waiting for more of an understanding of the impact of the work.

1. To better frame this piece I believe the manuscript would be more clearly a departure from the companion piece if the framing of the paper was towards understanding the inter-scenario responses vs. the inter-method responses in apparent temperature. This would also make the work more appropriate for submission to the special issue on solar geoengineering.

Reply: We developed a novel analysis of the role of SAI in air pollution from $PM_{2.5}$ part in our revised manuscript. This is the first such analysis made under geoengineering scenarios, and is of interest since $PM_{2.5}$ plays a serious role in health in the region.

2. I would encourage the authors to include more than apparent temperature in these results. Given this is a monsoon region is there a reason why precipitation was not a variable included with apparent temperature? Soil moisture is also a useful metric when understanding SAI; and could give the reader more insight into expected agricultural outcomes.

Reply: Apparent temperature is mainly related to the temperature, humidity and wind speed. In the formula of calculating AP, we can also know the relationship between AP and three meteorological factors. Precipitation itself has little impact on apparent temperature, which has little to do with whether it is located in the monsoon region. Crop yield is also an important index. Many studies show that the crop yield is projected to change under geoengineering scenarios, (Zhan et al., 2019; Fan et al., 2021). However, our main focus is the effect of geoengineering on human wellbeing. This is an expansive topic and we cannot deal completely with it in a single paper or reasonable length. So we chose to focus here on looking at changes that may occur as the climate is modified in the emissions and distribution of $PM_{2.5}$ aerosol. This is a novel analysis that could be of interest to a wide community.

References

Zhan, P., Zhu, W., Zhang, T., Cui, X., and Li, N.: Impacts of sulfate geoengineering on rice yield in China: Results from a multimodel ensemble, Earths Future, 7, 395-410, https://doi.org/10.1029/2018EF001094, 2019.

Fan, Y., Tjiputra, J., Muri, H., Lombardozzi, D., Park, C., Wu, S., and Keith, D.: Solar geoengineering can alleviate climate change pressures on crop yields, Nat. Food, 2, 373-381, https://doi.org/10.1038/s43016-021-00278-w, 2021.

3. The authors use AP>32 deg C as a metric citing that "similar differences between scenarios would apply for higher thresholds." I would be curious for the author to provide us with results based on a 32 deg C threshold as well as a higher limit instead of speculating here. This could also help the author tie these findings to tangible impacts, such as mortality, or even economic outcomes.

Reply: We cannot simply use any threshold because the less frequent the threshold the more statistically uncertain is the estimate of its probability. For example the well-known estimate for the uncertainty in an estimate of uncertainty, s, is $s/\sqrt{(2n-2)}$. So if we have only a very small number of instances of s, (that is n) then its uncertainty is very high. So we must compromise in having a measure of extreme that represents the tail of the distribution, while at the same time being common enough for a reasonable sampling of its likelihood in the 50 years or so of simulations available. This is why we choose NdAP_32 rather than say NdAP_27 or NdAP_39.

We revised the text:

This threshold does not lead to extreme risk and death, instead it is classified as requiring "extreme caution" by the US National Weather Service (National Weather Service Weather Forecast Office, *https://www.weather.gov/ama/heatindex*), but carries risks of heatstroke, cramps and exhaustion. A threshold of 39°C is classed as "dangerous" and risks heatstroke. While hotter AP thresholds would give a more direct estimate of health risks, the statistics of these presently rare events mean that detecting differences between scenarios is less reliable than using the cooler NdAP_32 threshold simply because the likelihood of rare events are more difficult to accurately quantify than more common events that are sampled more frequently. While there is evidence to suppose that in some distributions, the likelihood of extremes increases more rapidly than more central parts of a probability distribution – such as larger Atlantic hurricanes increasing faster than smaller ones (Grinsted et al., 2013), a conservative assumption is that similar differences between scenarios would apply for higher thresholds.

References

Grinsted, A., Moore, J., and Jevrejeva, S.: Projected Atlantic tropical cyclone threat from rising temperatures, PNAS, 110, 5369-5373, https://doi/10.1073/pnas.1209980110, 2013.

4. I would encourage the author to more clearly tie the variable changes to tangible impacts; specifically mortality or economic outcomes. Or at least make this a larger portion of the discussion. The results of the two downscaling methods employed can then provide a measure of uncertainty in the expected response to SAI or future warming.

Reply: We add the PM$_{2.5}$ part and discuss the contribution of changes in three variables to changes in PM$_{2.5}$ concentration.

Stylistically I found much of the results section difficult to decipher and was confused by qualitative descriptions of changes and the use of subjective adverbs. This section should be reconsidered with some rigor to provide the reader with a clearer quantitative description of each piece of analysis. I was also unable to see any equations in this manuscript – and based on the comments of reviewer 1 I would second concern regarding the use of linear regression to quantify contribution of wind, RH and T to the apparent temperature.

Reply: The wording has been changed where possible. This results in more repetitive wording.

The reason we used the regression approach is that this produces a least squares estimate of contributions. This is preferable in many statistical applications. However it may not be the best choice here as the assumptions of Normality and homoscedasticity in the analysis are probably not true. Using the referee #1's suggestions are more localized estimates around the mean values, which could be regarded as more statistically more robust, giving less weighting to outliers. But these are reasonable alternatives and the gradient or Jacobian approach plays a role in statistical analyses.

Please see the more detailed analysis we provide in response to Referee #1 on this issue. The conclusion we reached was:

The contributions of temperature and humidity are different using different methods, but the contribution of wind speed shows little change due to the linear relationship between wind speed and AP under either method. Changes in contribution from humidity is significant. In the referee's suggested method, the contribution of humidity is influenced by the hybrid effect of temperature, with big changes under higher temperature in JJA. As we all know, AP will change a lot under high temperature, although humidity changes little. In panel a and b, the contribution of humidity under referee's method is higher than that under previous method, but the opposite in the panel c and d. This is because different reference scenarios have different effects when calculating the contribution of humidity. For example, when we calculate the contribution of humidity on AP between G4 and RCP4.5, we can get the value of contribution A (we maintain the temperature in the G4 scenario and the humidity changes with the scenario) and B (we maintain the temperature in the RCP4.5 scenario and the humidity changes with the scenario), but A is not equal to B.

In summary, if we use the suggested method, the sum of changes in AP caused by three factors is not strictly equal to the absolute change in AP and the contribution of humidity and temperature is different when we select different reference between two scenarios. Actually, there is no best way to calculate contributions. Of course there are

uncertainties between different methods. We prefer our original method, so we retain it unchanged in our paper.

**Minor comments:**

(88) : I was confused by "Beijing experienced an increasing trend of 12.7%  or 2.07 days per decade in extreme warm nights (Wangetal.,2013) …" does this mean they experienced a percent increase of 12.07% per decade in the number of extremely warm nights – I was unable to confirm this based on the citation provided.

I was also just a bit confused why nights was the most useful metric here. Has anyone done a study of the increase in warm days from 1978-2008. It seems in line with your paper it could be useful to provide information regarding the historical increase in surface temperature vs. apparent temperature in this region.

Reply: We changed the sentence.

Over the period of 1971-2014, apparent temperature rises at a rate of 0.42°C/10 years over Beijing-Tianjin-Hebei region, with urbanization having an effect of 0.12°C/10 years (Luo and Lau, 2021).

(105) Define AP before using as an acronym

Reply: Done. We define the AP in line 45.

(109) Figure 1, Panels C and D color-bar labels should be added to specifiy units. I would also specific in the label what the red line is in figure 1, D – it seems that this is where the WRF domain terminates?

Reply: We pointed out in the annotation that the units of panels c and d refer to the number of people within the gird cell. The red line is the south boundary in WRF domain as is clear from panel b, and the that there are no blue cells south of the red line.

(127) Consider changing "climate forcing comes from 4 ESMs)" to something like climate simulations were performed by 4 ESMs – for clarity. As written it sounds like the radiative forcing from each model was extracted or somehow used separately.

Reply: We changed it:

Climate simulations are performed by 4 ESMs: BNU-ESM (Ji et al., 2014), HadGEM2-ES (Collins et al., 2011), MIROC-ESM (Watanabe et al., 2011) and MIROC-ESM-CHEM (Watanabe et al., 2011).

(151-158) I am unable to see any equations in the pdf preprint view of this document (I presume this is not an author issue but rather a technical issue!)

Reply: Ok. I am sure equations can be displayed normally.

(159-162) It would be useful to supply the read with values of the various thresholds for context as they read. I'm finding myself curious – what is the physiological maximum of the apparent temperature that humans can tolerate? What is the dangerous level? I would explain this before diving into your threshold value of 32 deg C to give the reader greater context. I would also provide citations of this empirically based scale.

Reply: We add the table in the supplementary information which gives the various thresholds and potential impacts according to by the US National Weather Service (National Weather Service Weather Forecast Office, *https://www.weather.gov/ama/heatindex*). We select the threshold AP of 32 as a trade off between rarity and hence uncertainty of its likelihood in each scenario, and the threat to health.

**Table S1**. Apparent temperature thresholds and its health impact (National Weather Service Weather Forecast Office, *https://www.weather.gov/ama/heatindex*).

| US NWS Classification | AP threshold | Effect on the body |
|---|---|---|
| Caution | 27-32°C | Prolonged exposure and/or physical activity can cause fatigue |
| Extreme caution | 32-39°C | Prolonged exposure and/or physical activity can lead to heatstroke, heat cramps, or heat exhaustion |
| Danger | 39-51°C | Heat cramps or heat exhaustion may occur, and prolonged exposure and/or physical activity may cause heatstroke |
| Extreme danger | >51°C | Very likely to suffer from heat stroke |

We add more text to contextualize the threshold.

This threshold does not lead to extreme risk and death, instead it is classified as requiring "extreme caution" by the US National Weather Service (National Weather Service Weather Forecast Office, *https://www.weather.gov/ama/heatindex*), but carries risks of heatstroke, cramps and exhaustion. A threshold of 39°C is classed as "dangerous" and risks heatstroke. While hotter AP thresholds would give a more direct estimate of health risks, the statistics of these presently rare events mean that detecting differences between scenarios is less reliable than using the cooler NdAP_32 threshold simply because the likelihood of rare events are more difficult to accurately quantify than more common events that are sampled more frequently. While there is evidence to suppose that in some distributions, the likelihood of extremes increases more rapidly than more central parts of a probability distribution – such as larger Atlantic hurricanes increasing faster than smaller ones (Grinsted et al., 2013), a conservative assumption is that similar differences between scenarios would apply for higher thresholds.

References

Grinsted, A., Moore, J., and Jevrejeva, S.: Projected Atlantic tropical cyclone threat from rising temperatures, PNAS, 110, 5369-5373, https://doi/10.1073/pnas.1209980110, 2013.

Figure 2: I would ask the author to revise the labeling of the terms – I am not following the utility of the bar chart here. These terms should be telling the reader information about the contribution to AP from each of three terms, however the bar chart makes it seem like terms 1-3 should add to the given AP; but on close inspection they do not? Unless my pdf view is not showing me, the coefficients on each term are also not defined directly in the text.

Reply: Maybe it's because the equation isn't shown in your downloaded preprint. In this figure, we show the equivalent temperature caused by three variables in Equation (1). The bars show the level of the equivalent temperature under each factor, and also shows the performance of downscaling on three variables.

Figure 3: I consider labeling the color bars of the bottom panels with # of days > 32 or something equivalent for clarity.

Reply: The color bar is labelled NdAP_32 and described in the caption as annual mean number of days with AP > 32°C

(234) : Missing "of" in the sentence "across most the North China.."

Reply: Done. We reedited this sentence.

Figure 4: consider modifying the titles of the lower plots for clarity to also read ISMIP and WRF, since all four plots are showing AP.

Reply: We have increased the font size of the title and widened the spacing between the top two panels and bottom two panels.

[Figure]

**Figure 4.** The probability density function (pdf) for daily apparent temperature under ISIMIP **(a, c)** and WRF **(b, d)** results in Beijing-Tianjin province **(a, b)** and Beijing-Tianjin urban areas **(c, d)** during 2008-2017.

Figure 5: The purpose of this figure is to compare the downscaling across WRF and ISIMIP – however the colorbar is constrained to give the reader cross scenario information. I would consider using different colorbars to for each scenario to better highlight differences in downscaling method, otherwise it seems these results cannot be well resolved by the reader.

Reply: Done.

[Figure]

**Figure 5.** Spatial pattern of ensemble mean apparent temperature difference (°C) under different scenarios over 2060-2069: G4-2010s (left column), G4-RCP4.5 (middle column) and G4-RCP8.5 (right column) based on ISIMIP and WRF methods. 2010s refers to the 2008-2017 period. Stippling indicates grid points where differences or changes are not significant at the 5% level according to the Wilcoxon signed rank test.

Also, this figure is called back in line 285 as Fig. 5a-5c; please add alphabetic denomination in the figure paneling.

Reply: I am sure that there are alphabetic denomination in each panel, maybe it is a display problem. In the new figure, I again added the alphabetic denomination.

(320-331): in expressing results in this section be more direct in statements and consider breaking into smaller sentences. Be more quantitative in these expressions and consider modifying the descriptor "significantly" as it is unclear if this means humidity is changed at a statistically significant amount in this context. In several sentences in this section the author quantifies the contribution of humidity or wind to changes in AP – make more clear to the reader what this percent is referring to (e.g. amount to over 3% of the total change in delta AP in summer).

Reply: We reedited this section.

Figure 7 shows the ISIMIP and WRF ensemble mean changes in the annual mean AP anomalies G4 during 2060-2069 relative to the past and the two future RCP scenarios. ISIMIP-downscaled AP (Fig. 7a-7c) shows significant anomalies (p<0.05) across the whole domain, even for the relatively small differences in G4-RCP4.5. ΔAP by WRF is lower than that by ISIMIP. Between G4 and 2010s, AP are projected to have increases of 1.8 (1.6), 2.1 (1.8), 2.4 (-0.2), 1.8 (0.8) °C from winter to autumn in ISIMIP (WRF) results. In ISIMIP results, the contribution of temperature ranges from 91%-104%, and

the contribution of wind speed ranges from 3%-10% in all seasons, while the contribution of humidity is negative or insignificant (Fig. 7a). However, the contribution of humidity is positive in WRF results (Fig. 7a). Between RCP4.5 and 2010s, annual mean AP is projected to increase by 3.0 °C and 1.8 °C in ISIMIP and WRF results respectively, which is higher than that between G4 and 2010s. The increase of temperature and decrease of wind speed have a significant impact on the annual average ΔAP contributed 97% (94%) and 4% (3%) in ISIMIP (WRF) results. The contributions of changes in humidity are significantly positive under G4 and RCP4.5 in WRF results, while it is the opposite in the ISIMIP results (Fig. 7a-7b).

Relative to RCP4.5 in the 2060s, AP is projected to decrease by 1.0 (0.4), 0.7 (0.8), 0.8 (0.7), and 1.3 (1.4) °C from winter to autumn under G4 in ISIMIP (WRF) results (Fig. 7c). In summer, the contribution from changes in temperature and humidity are 94% (105%) and 8% (-9%) in ISIMIP (WRF) results, respectively. There are insignificant contributions from wind speed under ISIMIP results, but a significant slight positive contribution (0.7%-4%) under WRF results (Fig. 7c). The annual mean AP under G4 is 2.8 (2.6) °C lower than that under RCP8.5 in ISIMIP (WRF) result. In this case, the contribution of changes in wind on ΔAP ranges from 3%-5% by ISIMIP, while it is close to 0 by WRF. As expected, ΔAP is mainly determined by the changes in temperature, with contributions usually above 90% between different scenarios.

Figure 8: Consider changing stippling to a cross grid hatching to allow the reader to perceive values in the bottom WRF panel that are not statistically significant but still provide context to the reader.

Reply: I reduced the size of the stippling points and updated the figure. The values which are covered by black points are nearly zero in the panel d-f.

[Figure]

**Figure 8.** Ensemble mean differences in annual number of days with AP > 32°C (NdAP_32) between scenarios for 2060-2069: G4-2010s (left column), G4-RCP4.5 (second column) and G4-RCP8.5 (right

column) based on ISIMIP method and WRF. 2010s means the results simulated during 2008-2017. Stippling indicates grid points where differences or changes are not significant at the 5% level according to the Wilcoxon signed rank test. Corresponding ISIMIP results for each ESM are in Fig. S11, and WRF results in Fig. S12.

(433) Change "warmer that 2m" to warmer than 2m.

Reply: Done.

AP is about 1.5°C warmer than 2 m temperature over the Beijing and Tianjin urban areas in summer.

---

## Referee Report (RR1)

This manuscript seeks to comprehend the changes in regional apparent temperature and PM2.5 concentrations under the conditions of global warming and sulfate aerosol injection. This understanding is achieved through the utilization of data from multiple Earth System Model simulations, two downscaling methods, and two statistic linear regression functions. The topic is both significant and innovative. Nevertheless, several substantial concerns persist:

The methodology employed to calculate PM2.5 concentration only considers factors such as temperature, humidity, wind speed, and anthropogenic emissions. However, two critical elements have been overlooked: precipitation, and natural aerosol emission. Precipitation has a crucial role in 'cleansing' air pollutants, including PM2.5, and future alterations in precipitation patterns could considerably influence regional PM2.5 concentrations. Furthermore, natural aerosol emissions, such as dust and sea salt, constitute more than half of the average global PM2.5. In regions like Beijing, "dust storms" are a significant air pollution phenomenon in the spring, contributing substantially to PM2.5 levels. The absence of these two factors from the calculation or discussion makes the projected future changes in PM2.5 unreliable.

Both apparent temperature and PM2.5 calculations use a simple linear regression. However, there exists a high correlation between the climate variables used, such as temperature and water vapor pressure/humidity. The uncertainties arising from this calculation method need to be addressed.

In the discussion section, the authors declare, "If we consider the aerosol deposition under G4 scenarios, PM2.5 concentration will be 0-1 µg/m3 higher than that without due to deposition of the SAI aerosols (Fig. S21)." This is incorrect. The injected sulfate aerosol would primarily deposit in the coarse mode and would not augment SO4 in PM2.5 compared to the reference case during the same period.

Lastly, the abstract lacks clarity in terms of the study's conclusions. How does PM2.5 change under future climate conditions and sulfate aerosol injection? What is the influence of the two downscaling methods on studying the health impact of SAI?

It is better to use climate intervention instead of geoengineering.

---

## Author Response (AR2)

Referee's comments are in red, our reply on black, quotes in the revised manuscript in blue.

**Referee#1**

I have limited my review to focus on areas which the authors have changed in response to my prior comments. Line numbers refer to the marked-up manuscript.

I was very impressed by the thoroughness of the authors' response. In particular I was happy to see the transition to an observational dataset in place of model reanalysis, and the extension to investigate $PM_{2.5}$ elevates the paper substantially and brings it much closer to publication. However this also means that the manuscript must now pass the same level of scrutiny as any other investigation of future $PM_{2.5}$, which is a high bar. This is the focus of my remaining concerns. The analysis is novel and I recognize the need for an efficient approach rather than (say) an additional set of CCM simulations, but the MLR approach used by the authors does cause me some concern. I have enumerated those concerns below and hope that the authors can address them.

The most significant issue is that a regression on a limited set of variables from historical data is used to predict future conditions. This is not inherently/fundamentally flawed, but there is a large body of literature which has investigated the nuanced relationship between future changes in climate and air quality, and how they are moderated by meteorological change (e.g. Jacob and Winner 2009, Fiore et al 2015). This has been looked at specifically in the context of health in China for ozone by e.g. Westervelt et al (2019). The challenge for modelers (including, now, the authors of this study) is whether past conditions accurately reflect the changes which will occur in the future. For example, it is possible that a geoengineering scenario could modify large scale dynamics in a way which is not reflected in past conditions, and which is different again from how those dynamics will be affected by climate change (Cheng et al 2022). It is also possible that the precursors dominating $PM_{2.5}$ will change, modifying the relationship between emissions and concentrations. Such changes would affect the patterns of pollution movement and evolution in a way which a local regression would not be able to capture. With that in mind, I would recommend three significant further revisions (two focused on the above and one on framing).

Reply: We would like to thank the referee for taking the time to review our manuscript again. Thank you very much for your affirmation of our first round of modification and constructive suggestions for the rationality of MLR in projecting $PM_{2.5}$. We have responded to the following comments one by one.

First, I recommend that the authors take an existing dataset of air quality outcomes for current and future conditions and show that the MLR method is capable of providing reasonable results when past conditions are used to build a regressor which predicts future $PM_{2.5}$ with evolving emissions and climate. One possibility in this regard would

be the AerChemMIP model outputs. It is fair to say that there is a lack of data to accomplish this for geoengineering output (although GeoMIP and/or GLENS output may be sufficient). If the authors can at least show that a regressor provides a reasonably accurate prediction under a significant change in climate and emissions that would significantly strengthen their findings in this paper.

Reply: We thank the referee's comments and suggestions. We found one paper which looked at the future $PM_{2.5}$ concentration in the similar region and asked for their data to assess our results. Li et al (2023) used the CMAQ model coupled WRF driven by GFDL-ESM2G and SMOKE model to explore the influence of emissions on air quality in the Beijing-Tianjin-Hebei region of China in 2050. The authors used the dynamical downscaled meteorological factors by WRF driven by GFDL-ESM2G and two air pollution emission scenarios, one is "base" based on the Beijing City Master Plan (2016-2035) and another is "EIT1" based on the emission reduction for WHO Interim Target-1 to compare the impact of different emission scenarios on $PM_{2.5}$ concentration in 2050 under RCP4.5. To assess the performance of our regression model we also downloaded the meteorological variables from GFDL-ESM2G under RCP4.5 and the "EIT1" emission data.

The statistical downscaled meteorological factors during 2008-2017 and 2050 under RCP4.5 were used as independent variables in the regression model to project $PM_{2.5}$ concentration in 2050 under RCP4.5 with the "EIT1" scenario. The spatial pattern is shown in the following figure S7. Although $PM_{2.5}$ concentration is nearly twice as high as from Li et al., $PM_{2.5}$ concentration from our regression model is also higher than the referenced data during 2008-2017, and our projections are similar to the spatial pattern of the seasonal $PM_{2.5}$ concentration from the chemical transport model, with correlation coefficient of 0.68-0.73. We also compare the spatial pattern of differences in $PM_{2.5}$ concentration between "base" and "EIT1" under RCP4.5 (Figure S8). Because of the small slope coefficient of $PM_{2.5}$ emission in our MLR we do not capture the large reduction of $PM_{2.5}$ concentration in the Beijing city center seen by Li et al (2023), (Fig. S8).

We added the following figures in the supplementary information.

[Figure]

**Figure S7.** Comparison of our MLR model projection and Li et al. (2023) RCP4.5 simulations. Li et al (2023) use the CMAQ model coupled WRF driven by GFDL-ESM2G and SMOKE model to

explore the influence of emissions on air quality in the Beijing-Tianjin-Hebei region of China in 2050. The authors used the dynamical downscaled meteorological factors by WRF driven by GFDL-ESM2G and two air pollution emission scenarios, one is "base" based on the Beijing City Master Plan (2016-2035) and another is "EIT1" based on the emission reduction for WHO Interim Target-1 to compare the impact of different emission scenarios on PM2.5 concentration in 2050 under RCP4.5. To assess the performance of our regression model we also downloaded the meteorological variables from GFDL-ESM2G under RCP4.5 and the "EIT1" emission data. The statistical downscaled meteorological factors during 2008-2017 and 2050 under RCP4.5 were used as independent variables in the regression model to project PM2.5 concentration in 2050 under RCP4.5 with the "EIT1" scenario. The top row are calculated by our regression model, and the bottom row are from Li et al. R is the correlation coefficient of PM$_{2.5}$ concentration spatial pattern between our results and Li et al.

[Figure]

**Figure S8.** Spatial pattern of differences in PM2.5 concentration under RCP4.5 between "base" and "EIT1" emission scenarios in Li et al (2023). The top row are calculated by our regression model, and the bottom row are from Li et al.

We added the following sentences in line 299.

We also tested the accuracy of our MLR model projection against simulations (Li et al., 2023) with the Community Multiscale Air Quality (CMAQ) model developed by the United States Environmental Protection Agency and which can simulate particulate matter on local scales (Foley et al., 2010; Yang et al., 2019) when coupled to WRF. We used the same meteorological forcing as Li with the "EIT1" PM$_{2.5}$ emissions scenario in 2050 under RCP4.5 (Fig. S7).

The spatial patterns are well correlated in all seasons (0.68-0.73), but PM$_{2.5}$ concentrations are about twice as high in our MLR model as from Li et al., (2023). PM$_{2.5}$ concentrations from our regression model are also higher than the referenced data during 2008-2017. While the difference in absolute PM2.5 concentrations are significant, we mainly consider differences of PM$_{2.5}$ concentration between G4 and RCP4.5/RCP8.5 in our study which we cannot compare these anomalies with the single

RCP4.5 scenario simulated by Li et al. (2023). We do compare the spatial pattern of differences in $PM_{2.5}$ concentration between "base" and "EIT1" under RCP4.5. Because of the small slope coefficient of $PM_{2.5}$ emission in our MLR, we do not capture the large reduction of $PM_{2.5}$ concentration in the Beijing city center seen by Li et al (2023), (Fig. S8).

References

Li, D., Wu, Q., Feng, J., Wang, Y., Wang, L., Xu, Q., Sun, Y., Cao, K., and Cheng, H.: The influence of anthropogenic emissions on air quality in Beijing-Tianjin-Hebei of China around 2050 under the future climate scenario, J. Cleaner Prod., 388, 135927, https://doi.org/10.1016/j.jclepro.2023.135927, 2023.

Foley, K. M., Roselle, S. J., Appel, K. W., Bhave, P. V., Pleim, J. E., Otte, T. L., Mathur, R., Sarwar, G., Young, J. O., Gilliam, R. C., Nolte, C. G., Kelly, J. T., Gilliland, A. B., and Bash, J. O.: Incremental testing of the Community Multiscale Air Quality (CMAQ) modeling system version 4.7, Geosci. Model Dev., 3, 205–226, https://doi.org/10.5194/gmd-3-205-2010, 2010.

Yang, X., Wu, Q., Zhao, R., Cheng, H., He, H., Ma, Q., Wang, L., and Luo, H.: New method for evaluating winter air quality: PM2.5 assessment using Community Multiscale Air Quality Modeling (CMAQ) in Xi'an, Atmos. Environ., 211, 18-28, https:// doi.org/10.1016/j.atmosenv.2019.04.019, 2019.

One of the most significant concerns I have in this respect is actually the nature of the regression. If I understand Sections 2.2 and 2.5 correctly, the authors are relating local $PM_{2.5}$ concentrations to local $PM_{2.5}$ emissions and local meteorology. However, $PM_{2.5}$ is will known to be influence by both upwind (i.e. regional) emissions of $PM_{2.5}$ and by emissions of $PM_{2.5}$ precursors such as SO2, NOx, and ammonia. The importance of taking these factors is elevated when looking at higher resolution data. Based on my interpretation of lines 285-293, these factors are not included in the MLR which would be concerning. If my interpretation is incorrect then I recommend that the authors clarify this in the relevant sections and specify clearly a) what precursors are considered, b) how spatial relationships between emissions (or other factors) and concentrations are captured, and c) how their model will be able to capture a shift in the chemical regime. Concerns a and c are only significant if secondary $PM_{2.5}$ is considered, so if instead only primary $PM_{2.5}$ is considered then I strongly recommend this be made very clear in the paper and the conclusions and abstract caveated appropriately. However, in either case the question regarding concern b remains.

Reply: Thank you very much for this comment. a) The reviewer is right, we only considered the primary $PM_{2.5}$ emissions and did not consider the precursor gases for secondary $PM_{2.5}$. Although secondary $PM_{2.5}$ emission is not included, $PM_{2.5}$ concentration includes both primary and secondary $PM_{2.5}$ in our model. b) The referee

is correct that $PM_{2.5}$ concentration is not only related to local meteorological conditions and emissions. Limited by our model being a statistical model rather than a chemical transport model, we expect that by having meridional and latitudinal winds as variables in our model that these $PM_{2.5}$ advections can be accounted for. c) We note that the future precursor mix will change in ways that are rather speculative as they depend on technological innovation and policies that are inherently unpredictable.

We have added the following sentences in line 275.

Here, we use $PM_{2.5}$ concentration including both primary and secondary $PM_{2.5}$ as the dependent variable and primary $PM_{2.5}$ emission and meteorological factors as independent variables in the MLR. Future $PM_{2.5}$ emissions will change in ways that are rather speculative as they depend on technological innovation and policies that are inherently unpredictable. The MLR assumes that the past emissions mix and secondary aerosols remain unchanged in the future, but meteorological factors will also indirectly impact secondary $PM_{2.5}$ to some extent.

We have added the following sentences in line 810.

Our study did not consider the impacts of socio-economic pathways on $PM_{2.5}$ future emissions, instead we explore the meteorological differences between the SAI G4 scenario and the greenhouse gas RCP4.5/RCP8.5 on $PM_{2.5}$ concentrations. $PM_{2.5}$ emissions were defined by the uncontrolled ("baseline") and a scenario where technological intervention ("mitigation") reduces emissions. There are some limitations in our study. Firstly, the HTAP_V3 dataset only includes anthropogenic $PM_{2.5}$ emission, not natural $PM_{2.5}$ emission. Natural $PM_{2.5}$ will also change in the future under changing climate. The sources of natural $PM_{2.5}$ include the sandstorms that sometimes occur in spring as extreme winds mobilize dry unvegetated soils. These relatively extreme conditions are difficult to simulate in ESM and subject to land use policy e.g., the numerous ecosystem service measures undertaken by China over the last five decades (Miao et al.,2015). Secondly, although $PM_{2.5}$ concentration includes both primary and secondary $PM_{2.5}$ during model training, we do not consider the precursor gases for secondary $PM_{2.5}$ directly. The sensitivity of MLR may diminish at the high $PM_{2.5}$ values when secondary $PM_{2.5}$ dominates the variability of total $PM_{2.5}$ (Upadhyay et al., 2018). Thirdly, we only consider the effect of dominant near-surface meteorological variables on the $PM_{2.5}$. However, the vertical transport of pollutants related to vertical atmospheric stability should not be ignored (Lo et al., 2006; Wu et al., 2005), and this may contribute to the differences in RCP4.5 scenario from our MLR model and more sophisticated simulations (Fig. S7). Finally, although it is insignificant for the Beijing and Tianjin provinces, the MLR model suffers collinearity problems in some areas. These factors play smaller roles as we are mainly considering changes in $PM_{2.5}$ concentration between different climate scenarios. Nevertheless, projection for changes in $PM_{2.5}$ between SAI scenarios and per greenhouse gas scenarios would be valuable for global air quality impacts from geoengineering.

References

Lo, J., Lau, A., Fung, J., and Chen, F.: Investigation of enhanced cross-city transport and trapping of air pollutants by coastal and urban land-sea breeze circulations, J. Geophys. Res.-Atmos., 111(D14), https://doi.org/10.1029/2005JD006837, 2006.

Wu, D., Tie, X., Li, C., Ying, Z., Kai-Hon Lau, A., Huang, J., Deng, X., and Bi, X.: An extremely low visibility event over the Guangzhou region: a case study, Atmos. Environ., 39, 6568-6577, https://doi.org/10.1016/j.atmosenv.2005.07.061, 2005.

Miao, L., Moore, J. C., Zeng, F., Lei, J., Ding, J., He, B., and Cui, X.: Footprint of research in desertification management in China, Land Degrad. Dev., 26, 450-457, https://doi.org/10.1002/ldr.2399, 2015.

Second, I recommend that the authors compare their findings against existing projections of the change in surface $PM_{2.5}$ in the target region over the next 40 years. There are several studies looking at how surface air quality in China might evolve under different scenarios (see e.g. Hong et al 2019). Showing that the regression-based approach can recover the majority of the climate change-induced signal would be valuable not only from the perspective of this paper, but from the perspective of the field more broadly.

Reply: Thank you, we added some sentences in the discussion in line 765.

Xu et al. (2021) projected 2030 $PM_{2.5}$ concentrations will decrease by 8.8% and 5.5% under RCP4.5 and RCP8.5 respectively relative to 2015. Wang et al. (2021) also projected decreasing trends in China under RCP4.5 and RCP8.5 during 2030-2050. There were seasonal changes in $PM_{2.5}$ concentration differences between RCP4.5/8.5 scenarios and the historical scenario near the Bohai Sea (Dou et al., 2021). However, there are also some simulations where $PM_{2.5}$ concentrations increase in warmer climates. Hong et al. (2019) suggest that annual mean $PM_{2.5}$ concentrations will increase 1-8 $\mu g/m^3$ in an area including Beijing and Tianjin under RCP4.5 during 2046-2050, compared with 2006-2010. These inconsistent responses are mainly caused by the differences in the selection of ESMs, chemical transport models and climate/emission scenarios. Different RCP scenarios not only correspond to different future climate states, but also have different anthropogenic emissions of air pollutants. In our study, we do not consider the $PM_{2.5}$ emission differences between RCP4.5 and RCP8.5, and instead applied the ECLIPSE $PM_{2.5}$ emission scenarios in our MLR projection.

References

Xu, J., Yao, M., Wu, W., Qiao, X., Zhang, H., Wang, P., Yang, X., Zhao, X., and Zhang, J.: Estimation of ambient $PM_{2.5}$-related mortality burden in China by 2030 under climate and population change scenarios: A modeling study, Environ, Int., 156,106733, https://doi.org/10.1016/j.envint.2021.106733, 2021.

Wang, Y., Hu, J., Zhu, J., Li, J., Qin, M., Liao, H., Chen, K., and Wang, M.: Health Burden and economic impacts attributed to $PM_{2.5}$ and $O_3$ in China from 2010 to 2050 under different representative concentration pathway scenarios, Resour. Conserv. Recy., 173, 105731, https://doi.org/10.1016/j.resconrec.2021.105731, 2021.

Dou, C., Ji, Z., Xiao, Y., Zhu, X., and Dong, W.: Projections of air pollution in northern China in the two RCPs scenarios, Remote Sens., 13, 3064, https://doi.org/10.3390/rs13163064, 2021.

Hong, C., Zhang, Q., Zhang, Y., Davis, S., Tong, D., Zheng, Y., Liu, Z., Guan, D., He, K., and Schellnhuber, H. J.: Impacts of climate change on future air quality and human health in China, PNAS, 116, 17193-17200, https://doi.org/10.1073/pnas.1812881116, 2019.

Finally, I am surprised that the abstract and conclusions still do not provide any quantitative data regarding how the different downscaling methods affect the outcomes inspected here. By including the extension to $PM_{2.5}$ I think the authors have done a good job of addressing my prior major concern (of this manuscript having no novelty when considered next to their existing work), but it would be helpful to include some high level conclusions regarding the degree to which model- (WRF) or statistics-based (ISIMIP) downscaling results in different or similar outcomes for health risks under different scenarios.

Reply: Thanks for your suggestions. we add some sentences in line 781 in the discussion.

There are some differences in projecting $PM_{2.5}$ concentration between WRF and ISIMIP methods. Compared to the 2010s reference, $PM_{2.5}$ concentration in ISIMIP are projected to decrease more than using WRF in G4 under the "mitigation" scenario during the 2060s over the Tianjin province (Fig. 11a, e). However, the spatial patterns of changes in $PM_{2.5}$ concentration between G4 and RCP4.5/8.5 under the "mitigation" scenario during 2060s are similar (Fig. 11c-d, g-h). This means that the effects of different downscaled methods on projecting $PM_{2.5}$ are small if we only consider the climate change alone without considering emissions changes. Due to the larger regression coefficient of emissions in the MLR under the ISIMIP method (Fig. S25, S26), the negative changes in $PM_{2.5}$ concentration are larger between "mitigation" and baseline under G4 during 2060s than that under the WRF method. Correspondingly, the ISIMIP method has a greater reduction in $PM_{2.5}$ related RR than WRF under three future climate scenarios during the 2060s.

We add the following sentences in line 29 in the abstract.

Compared with the 2010s, $PM_{2.5}$ concentration is projected to decrease 5.4 μg/m$^3$ in the Beijing-Tianjin province under the G4 scenario during the 2060s from the WRF downscaling, but decrease by 7.6 μg/m$^3$ using ISIMIP. The relative risk of 5 diseases decreases by 1.1%-6.7% in G4/RCP4.5/RCP8.5 using ISIMIP, but have smaller decrease (0.7%-5.2%) using WRF.

Minor comments

While I understand the authors' statement that health impacts only matter when people are affected, I still believe that line 270 ("Since health impacts are more important where there are more people") is likely to cause misunderstanding. I would recommend wording instead along the lines of "Since health impacts scale with the number of people affected". As written, it sounds like a single person's exposure is more important if they live in an urban rather than rural environment, when the intended meaning is instead (presumably) that an increase in concentration causes more health impact when a large number of people are exposed.

Reply: Done. We have changed the original sentence with that you suggested.

Since health impacts scale with the number of people affected,

Upon review, it appears that the Eastham et al. (2018) study does include limited meteorological effects (line 120). It would perhaps be more accurate to state that the study included only a first-order estimate of temperature and precipitation change.

Reply: Done. We rewrote the sentence in line 114.

However, this study included only a first-order estimate of temperature and precipitation change on $PM_{2.5}$ concentration under geoengineering, and also did not consider the situation in a highly polluted urban environment such as included in our domain, and which is typical of much of the developing world.

There remain some minor grammar and spelling errors (e.g. "statistically approach" on line 21, "gird" on line 334, "includes" should be "include" on line 326). Similarly, there is some confusing wording (e.g. "the ~1-2% wetter humidity has ~10% negative effect on decrease of $PM_{2.5}$" – the multiple negatives here make it difficult to understand whether increasing humidity is causing an increase or decrease in $PM_{2.5}$, "2.5" in $PM_{2.5}$ is not subscripted in line 944). These are rare but I would suggest the authors take

another pass through the manuscript to clean up these few issues.

Reply: Done. We apologize for our errors, and we have rewritten the sentence in line 623 and make it clear. We also corrected all the subscripts of $PM_{2.5}$ in the manuscript.

The ~1-2% increase of humidity leads to ~10% increase of $PM_{2.5}$ concentration in the south of Beijing (Fig. 12g), and 0.2-0.3 m/s deceases of U-wind leads to 0-10% increase of $PM_{2.5}$ concentration in Zhangjiakou (Fig. 12h).

References

Cheng, W., MacMartin, D. G., Kravitz, B., Visioni, D., Bednarz, E. M., Xu, Y., Luo, Y., Huang, L., Hu, Y., Staten, P. W., Hitchcock, P., Moore, J. C., Guo, A., and Deng, X.: Changes in Hadley circulation and intertropical convergence zone under strategic stratospheric aerosol geoengineering, npj Climate and Atmospheric Science, 5, 1–11, 2022.

Fiore, A. M., Naik, V., and Leibensperger, E. M.: Air quality and climate connections, J. Air Waste Manag. Assoc., 65, 645–685, 2015.

Hong, C., Zhang, Q., Zhang, Y., and Schellnhuber, H. J.: Impacts of climate change on future air quality and human health in China, PNAS, 2019.

Jacob, D. J. and Winner, D. a.: Effect of climate change on air quality, Atmos. Environ., 43, 51–63, 2009.

Westervelt, D. M., Ma, C. T., He, M. Z., Fiore, A. M., Kinney, P. L., Kioumourtzoglou, M.-A., Wang, S., Xing, J., Ding, D., and Correa, G.: Mid-21st century ozone air quality and health burden in China under emissions scenarios and climate change, Environ. Res. Lett., 14, 074030, 2019.

**Referee#2**

This manuscript seeks to comprehend the changes in regional apparent temperature and $PM_{2.5}$ concentrations under the conditions of global warming and sulfate aerosol injection. This understanding is achieved through the utilization of data from multiple Earth System Model simulations, two downscaling methods, and two statistic linear regression functions. The topic is both significant and innovative. Nevertheless, several substantial concerns persist:

Reply: We would like to thank the referee for taking the time to review our manuscript again. Thanks for your positive response and constructive comments for our new manuscript. We have responded to your comments one by one.

The methodology employed to calculate $PM_{2.5}$ concentration only considers factors such as temperature, humidity, wind speed, and anthropogenic emissions. However, two critical elements have been overlooked: precipitation, and natural aerosol emission. Precipitation has a crucial role in 'cleansing' air pollutants, including $PM_{2.5}$, and future alterations in precipitation patterns could considerably influence regional $PM_{2.5}$ concentrations. Furthermore, natural aerosol emissions, such as dust and sea salt, constitute more than half of the average global $PM_{2.5}$. In regions like Beijing, "dust storms" are a significant air pollution phenomenon in the spring, contributing substantially to $PM_{2.5}$ levels. The absence of these two factors from the calculation or discussion makes the projected future changes in $PM_{2.5}$ unreliable.

Reply: We agree with the referee's concern of variables in our regression model. Actually, multiple meteorological factors are contributed to $PM_{2.5}$ concentration, such as temperature (You et al., 2017), humidity (Cheng et al., 2017), wind (Yin et al., 2017), precipitation (Guo et al., 2016), atmospheric pressure (Zhang et al., 2015), radiation (Chen et al., 2017) and planetary boundary layer height (Zheng et al., 2017) etc. Crudely, the dominant meteorological factors vary with areas. In our analysis, we did not apply all possible variables in our regression model, and we only considered the main meteorological factors in our domain. Chen et al (2020) pointed out that humidity and wind speed are the two dominant meteorological factors in the Jing-Jin-Ji region (which contains Beijing and Tianjin). Based on their study, we included temperature, humidity, as well as meridional and latitudinal winds into our regression model. Natural emission, such as "dust storms", also contributed to $PM_{2.5}$ concentration, but the composition of dust is complex, generally in Beijing bringing in coarser $PM_{10}$ and not so much $PM_{2.5.}$ Furthermore, the sources of natural $PM_{2.5}$ include the sandstorms that sometimes occur usually in spring as extreme winds mobilize dry unvegetated soils. These extreme conditions are difficult to simulate in ESM and subject to land use policy e.g. the numerous ecosystem service measures undertaken by China over the last five decades (Miao et al.,2015). On the other hand, both HTAP_V3 and ECLIPSE V6b dataset do not offer the natural aerosol emission. So, there are some limitations in our study. In regard to precipitation, the ESM estimates of anomalies relative to historical are 24.0,

45.3, and 63.2 mm/year under G4, RCP4.5 and RCP8.5, respectively (Table S2). Among the four ESMs, no ESM shows significant changes, although differences are significant for the ensemble mean between RCP8.5 and 2010s.

We add some sentences in our manuscript.

We add the following sentences in line 260.

Many meteorological factors, such as temperature (You et al., 2017), precipitation (Guo et al., 2016), wind speed (Yin et al., 2017), radiation (Chen et al., 2017), planetary boundary layer height (Zheng et al., 2017) etc., can affect the $PM_{2.5}$ concentration. Their relative importance differs regionally. But here we consider only differences that are produced by the three scenarios, so for example we do not include precipitation in our analysis because none of the ESM simulate significant changes in our domain (Table S2).

We add the following sentences in line 810.

Our study did not consider the impacts of socio-economic pathways on $PM_{2.5}$ future emissions, instead we explore the meteorological differences between the SAI G4 scenario and the greenhouse gas RCP4.5/RCP8.5 on $PM_{2.5}$ concentrations. $PM_{2.5}$ emissions were defined by the uncontrolled ("baseline") and a scenario where technological intervention ("mitigation") reduces emissions. There are some limitations in our study. Firstly, the HTAP_V3 dataset only includes anthropogenic $PM_{2.5}$ emission, not natural $PM_{2.5}$ emission. Natural $PM_{2.5}$ will also change in the future under changing climate. The sources of natural $PM_{2.5}$ include the sandstorms that sometimes occur in spring as extreme winds mobilize dry unvegetated soils. These relatively extreme conditions are difficult to simulate in ESM and subject to land use policy e.g., the numerous ecosystem service measures undertaken by China over the last five decades (Miao et al.,2015). Secondly, although $PM_{2.5}$ concentration includes both primary and secondary $PM_{2.5}$ during model training, we do not consider the precursor gases for secondary $PM_{2.5}$ directly. The sensitivity of MLR may diminish at the high $PM_{2.5}$ values when secondary $PM_{2.5}$ dominates the variability of total $PM_{2.5}$ (Upadhyay et al., 2018). Thirdly, we only consider the effect of dominant near-surface meteorological variables on the $PM_{2.5}$. However, the vertical transport of pollutants related to vertical atmospheric stability should not be ignored (Lo et al., 2006; Wu et al., 2005), and this may contribute to the differences in RCP4.5 scenario from our MLR model and more sophisticated simulations (Fig. S7). Finally, although it is insignificant for the Beijing and Tianjin provinces, the MLR model suffers collinearity problems in some areas. These factors play smaller roles as we are mainly considering changes in $PM_{2.5}$ concentration between different climate scenarios. Nevertheless, projection for changes in $PM_{2.5}$ between SAI scenarios and per greenhouse gas scenarios would be valuable for global air quality impacts from geoengineering.

We add the following table in the supplementary information.

**Table S2.** The changes in annual mean precipitation (mm/year) between G4/RCP4.5/RCP8.5 during 2060s and references during 2010s over the domain. Bold indicates that differences are significant.

|  | G4-2010s | RCP4.5-2010s | RCP8.5-2010s |
|---|---|---|---|
| MIROC-ESM | 73.1 | 50.5 | 51.8 |
| MIROC-ESM-CHEM | -4.9 | 43.2 | 47.1 |
| HadGEM2-ES | 69.1 | 114.1 | 147.6 |
| BNU-ESM | -41.6 | -26.6 | 6.4 |
| Ensemble | 24.0 | 45.3 | **63.2** |

References

You, T., Wu, R., Huang, G., Fan, G.: Regional meteorological patterns for heavy pollution events in Beijing, J. Meteorolog. Res., 31, 597-611, https://doi.org/10.1007/s13351-017-6143-1, 2017.

Cheng, L., F, M., Chen, L., Jiang, T., Su, L.: Effects on the haze pollution from autumn crop residue burning over the Jing-Jin-Ji Region, China Environ. Sci., 37, 2801-2812, 2017.

Yin, Z., Wang, H., and Chen, H.: Understanding severe winter haze events in the North China Plain in 2014: roles of climate anomalies, Atmos. Chem. Phys., 17, 1641–1651, https://doi.org/10.5194/acp-17-1641-2017, 2017.

Guo, L., Zhang, Y., Lin, H., Zeng, W., Liu, T., Xiao, J., Rutherford, S., You, J., Ma, W.: The washout effects of rainfall on atmospheric particulate pollution in two Chinese cities, Environ. Pollut., 215, 195-202, https://doi.org/10.1016/j.envpol.2016.05.003, 2016.

Zhang, Y., Cao, F.: Fine particulate matter ($PM_{2.5}$) in china at a city level, Sci. Rep., 5, 14884, https://doi.org/10.1038/srep14884, 2015.

Chen, Z., Cai, J., Gao, B., Xu, B., Dai, S., He, B., Xie, X.: Detecting the causality influence of individual meteorological factors on local $PM_{2.5}$ concentrations in the Jing-Jin-Ji region, Sci. Rep., 7, 40735, https://doi.org/10.1038/srep40735, 2017.

Zheng, C., Zhao, C., Zhu, Y., Wang, Y., Shi, X., Wu, X., Chen, T., Wu, F., and Qiu, Y.: Analysis of influential factors for the relationship between $PM_{2.5}$ and AOD in Beijing, Atmos. Chem. Phys., 17, 13473–13489, https://doi.org/10.5194/acp-17-13473-2017, 2017.

Both apparent temperature and $PM_{2.5}$ calculations use a simple linear regression.

However, there exists a high correlation between the climate variables used, such as temperature and water vapor pressure/humidity. The uncertainties arising from this calculation method need to be addressed.

Reply: Yes, collinearity of variables is inevitable in our domain. The domination of the seasonal winter and summer monsoonal weather patterns mean that temperatures, precipitation and wind direction are all highly seasonal and correlated. In winter, precipitation is minimal and northerly winds predominate, in summer the opposite is true. However, the three fields are important in their own right since emission sources are essentially absent from the north, while temperature and humidity dominate aerosol microphysics. Furthermore, we used a widely used empirical formula to calculate the apparent temperature (Steadman 1984), that combines various meteorological fields.

We use the variance inflation factor (VIF) to test if there is excessive collinearity in our MLR model. Generally, if VIF value is greater than 10, there is collinearity problem between variables. As shown in figure S3 below, there are indeed collinearity problems in some areas. The problem doesn't occur in Beijing-Tianjin province, so there is no impact on the results for Beijing-Tianjin urban areas. To further explore the impact of collinearity on the results in high VIF grid cells, we compared the differences of $PM_{2.5}$ concentration in the future between removing factors with VIF greater than 10 and the full variables model (figure S4 and figure S5). Using ISIMIP downscaling, we only removed the temperature, while we removed the temperature and U-wind in the WRF method. In figure S4, we can see that $PM_{2.5}$ concentration show an increase of ~1 ug/m$^2$ in all ESMs under G4 with the "baseline" scenario (except HadGEM2-ES under ISIMIP method) after dealing the collinearity problem. In figure S5, $PM_{2.5}$ concentration has nearly 5-15 ug/m$^2$ decrease in all ESMs under G4 with "mitigation" scenario after dealing the collinearity problem. This means that $PM_{2.5}$ concentration has more sensitivity to the $PM_{2.5}$ emission after dealing the collinearity problem. The difference in $PM_{2.5}$ concentration between different scenarios with the removal of collinearity variables is shown in the following Figure 11, and that without removal of collinearity is shown in the Figure S18. The reductions in $PM_{2.5}$ between G4 and 2010s are a little higher in the area where there are collinearity problems after dealing the collinearity problem.

Although the absolute $PM_{2.5}$ concentration is different whether we consider collinearity or not, there are little differences in the changes of $PM_{2.5}$ concentration between G4 and 2010s/RCP4.5/RCP8.5. We also acknowledge that there are large uncertainties in $PM_{2.5}$ concentration in the future with considering collinearity or not. But in our study, we pay attention to the differences of $PM_{2.5}$ concentration between G4 and RCP4.5/8.5. So considering collinearity is not so important, and as shown there is no collinearity problem in Beijing-Tian province.
We rewrote the sentences in line 223.

We used a widely used empirical formula to calculate the apparent temperature

(Steadman 1984), that combines various meteorological fields, which also has been widely used to study heat waves, heat stress and temperature-related mortality.

We add the following sentences in line 281.

Collinearity of variables is inevitable in our domain. The domination of the seasonal winter and summer monsoonal weather patterns mean that temperatures, precipitation and wind direction are all highly seasonal and correlated. In winter, precipitation is minimal and northerly winds predominate, in summer the opposite is true. These three meteorological fields are important also important for emissions, since sources are essentially absent from the north, while temperature and humidity dominate aerosol microphysics.

We use the variance inflation factor (VIF) to test if there is excessive collinearity in our MLR models. Generally, if VIF value is greater than 10, there is collinearity problem between variables. Figure S3 shows that there are indeed collinearity problems in some areas, but not in Beijing-Tianjin province, so there is no impact on the results for the urban areas. We explored the impact of collinearity on the results in high VIF grid cells by removing factors with VIF greater than 10 and the full variables model (Fig. S4 and Fig. S5). Using ISIMIP downscaling, we only removed the temperature, while we removed the temperature and U-wind in the WRF method. PM$_{2.5}$ concentrations increased by ~1 µg/m$^2$ in all ESMs under G4 with the "baseline" scenario (Fig. S4), in contrast, PM$_{2.5}$ concentrations decreased by 5-15 µg/m$^2$ with the "mitigation" scenario (Fig. S5) after dealing the collinearity problem. This means that PM$_{2.5}$ concentration has more sensitivity to the PM$_{2.5}$ emission after accounting for collinearity. Although the absolute PM$_{2.5}$ concentrations are different accounting for collinearity, there are no significant differences in the changes of PM$_{2.5}$ concentration between G4 and the 2010s/RCP4.5/RCP8.5 (Fig.11, Fig. S18).

We reploted the following figures in the manuscript.

[Figure]

**Figure 11.** Spatial patterns of ensemble mean PM$_{2.5}$ concentration difference (µg/m$^3$) between

"mitigation" under G4 in the 2060s and reference **(a, e)**, between "mitigation" and "baseline" under G4 in the 2060s **(b, f)**, between G4 and RCP4.5 under "mitigation" scenario in the 2060s **(c, g)**, and between G4 and RCP8.5 under "mitigation" scenario in the 2060s **(d, h)** based on ISIMIP **(a-d)** and WRF **(e-h)** results. Excessive collinearity variables have been removed (Fig. S18 shows the results without this procedure). Stippling indicates grid points where differences or changes are not significant at the 5% significant level according to the Wilcoxon signed rank test.

We added the following figures in the supplementary information.

[Figure]

**Figure S3.** Variance inflation factor (VIF) test of excessive collinearity in our MLR model. VIF >10 means there is collinearity problem between variables (dotted regions).

[Figure]

**Figure S4.** Difference in PM$_{2.5}$ concentration under G4 with "baseline" scenario in 2060s between removing factors with VIF greater than 10 and the full variables model.

[Figure]

**Figure S5.** Difference in PM$_{2.5}$ concentration under G4 with "mitigation" scenario in 2060s between removing factors with VIF greater than 10 and the full variables model.

[Figure]

**Figure S18**. Same as figure 11, but the results of all variables in MLR.

We updated Fig.2, Fig.11-14, Table S2, Table S3, Fig.S3, Fig.S13-S14 and Fig. S19-S21 after removing the collinearity variables in the areas with VIF>10 in the original unrevised manuscript. We have also revised the sentences in the manuscript and the numbering of figures accordingly, and overall, the changes are not very significant.

In the discussion section, the authors declare, "If we consider the aerosol deposition under G4 scenarios, PM$_{2.5}$ concentration will be 0-1 μg/m3 higher than that without due to deposition of the SAI aerosols (Fig. S21)." This is incorrect. The injected sulfate aerosol would primarily deposit in the coarse mode and would not augment SO$_4$ in PM$_{2.5}$ compared to the reference case during the same period.

Reply: The referee gives no support for the assertion that the numbers we calculate are incorrect. This concerns the deposition from the SAI as PM$_{2.5}$. Eastham et al. (2018) considered this with a much more sophisticated treatment than available to us. They

concluded that 1/25 of the SAI was deposited as $PM_{2.5}$. This is the ratio we use, and since it is the only study to simulate the effects, we will continue to use this number.

Lastly, the abstract lacks clarity in terms of the study's conclusions. How does $PM_{2.5}$ change under future climate conditions and sulfate aerosol injection? What is the influence of the two downscaling methods on studying the health impact of SAI?
It is better to use climate intervention instead of geoengineering.

Reply: Thanks for your comments. We changed the stratospheric aerosol injection to stratospheric aerosol intervention. We are limited in the number of words in the abstract. We add some sentences in the abstract.

Compared with the 2010s, $PM_{2.5}$ concentration is projected to decrease 5.4 $\mu g/m^3$ in the Beijing-Tianjin province under the G4 scenario during the 2060s from the WRF downscaling, but decrease by 7.6 $\mu g/m^3$ using ISIMIP. The relative risk of 5 diseases decreases by 1.1%-6.7% in G4/RCP4.5/RCP8.5 using ISIMIP, but have smaller decrease (0.7%-5.2%) using WRF.